

# Numerical Modeling of Fluid Effects on Seismic Properties of Fractured Magmatic Geothermal Reservoirs

Melchior Grab[1], Beatriz Quintal[2], Eva Caspari[2], Hansruedi Maurer[1], and Stewart Greenhalgh[3]

[1]Institute of Geophysics, ETH Zürich, Zürich 8092, Switzerland
[2]Institute of Earth Science, University of Lausanne, Lausanne 1015, Switzerland
[3]Department of Geosciences, King Fahd University of Petroleum & Minerals, Dhahran 31261, Saudi Arabia

*Correspondence to:* Melchior Grab (melchior.grab@erdw.ethz.ch)

**Abstract.** Seismic investigations of geothermal reservoirs over the last 20 years have sought to interpret the resulting tomograms and reflection images in terms of the degree of reservoir fracturing and fluid content. Since the former provides the pathways and the latter acts as the medium for transporting geothermal energy, such information is needed to evaluate the quality of the reservoir. In conventional rock physics-based interpretations, this hydro-mechanical information is approximated
from seismic velocities computed at the low frequency (field-based) and high frequency (lab-based) limits. In this paper, we demonstrate how seismic properties of fluid-filled, fractured reservoirs can be modeled over the full frequency spectrum using a numerical simulation technique which has become popular in recent years. It is based on Biot's theory of poroelasticity and enables the modeling of the seismic velocity dispersion and the frequency dependent seismic attenuation due to wave-induced fluid flow. These properties are sensitive to key parameters such as the hydraulic permeability of fractures as well
as the compressibility and viscosity of the pore fluids. Applying the poroelastic modeling technique to the specific case of a magmatic geothermal system under stress due to the weight of the overlying rocks requires careful parameterization of the model. This includes consideration of the diversity of rock types occurring in the magmatic system and examination of the confining pressure-dependency of each input parameter. After the evaluation of all input parameters, we use our modeling technique to determine the seismic attenuation factors and phase velocities of a rock containing a complex interconnected fracture network, whose geometry is based on a fractured geothermal reservoir in Iceland. Our results indicate that in a magmatic
geothermal reservoir the overall seismic velocity structure mainly reflects the lithological heterogeneity of the system, whereas indicators for reservoir permeability and fluid content are deducible from the magnitude of seismic attenuation and the critical frequency at which the peak of attenuation and maximum velocity dispersion occur. The study demonstrates how numerical modeling provides a valuable tool to overcome interpretation ambiguity and to gain a better understanding of the hydrology of geothermal systems, which are embedded in a highly heterogeneous host medium.




# 1 Introduction

Magmatic geothermal reservoirs consist of permeable extrusive and intrusive rock formations, situated at depths where sufficiently high temperatures prevail. They are saturated with hot fluids, and usually heated by magma intrusions beneath the system. Evaluating the quality of such a reservoir requires an estimate of the fluid enthalpy and of the host rock permeability. Seismic methods are amongst the most efficient exploration techniques to image the deep subsurface. The key quantities which can be obtained from a seismic survey are the geometry of subsurface interfaces (e.g. lithological boundaries, faults, fracture zones), the P- and S-wavespeeds ($V_P$ and $V_S$) of various rock units, and the corresponding seismic attenuation characteristics. The latter is expressed by the inverse of the P- and S-wave specific quality factors $Q_P$ and $Q_S$. The challenge in seismic interpretation is to link these seismic properties with the geological/hydrological properties of interest.

To constrain the seismic interpretation, it is recommended to measure the elastic and anelastic rock properties of small rock specimens in the laboratory under in situ pressure-, temperature-, and fluid content-conditions. However, in magmatic geothermal systems, the host rock is often highly impermeable and the fluid transport predominately takes place within macro-fracture networks, rather than through the matrix. Such fractures are not present in the rock samples investigated in the laboratory, due to their limited size. Therefore, laboratory experiments only provide the properties of relatively intact rock and indicators for the presence or absence of fluids need to be deduced from fluid-rock interactions at larger scales through rock physics concepts. Various such concepts of differing complexity have been used over the last 20 years to interpret seismic tomograms from geothermal exploration campaigns in magmatic environments.

Perhaps the simplest and most straightforward way to infer the presence of fluids in seismic interpretation is to recognize that $V_P$ is more sensitive to fluid saturation effects than $V_S$, as the presence of liquids tends to increase $V_P$ but not significantly change $V_S$. Thus it is common practice to deduce fluid saturation from seismic tomograms by interpreting the $V_P/V_S$ ratio in a qualitative manner. For instance Sanders et al. (1995) and Jousset et al. (2011) interpreted $V_P/V_S$ anomalies to be indicative of the presence of supercritical fluids in a formation of the geothermal system in the Long Valley Caldera, California and in the Hengill volcanic complex in Iceland, respectively. Gunasekera et al. (2003), who conducted a time-lapse local earthquake tomography study in The Geysers, California, over a time period of 7 years, interpreted temporal variations in a $V_P/V_S$ anomaly during the time of observation as an indication of water depletion resulting from reservoir operation.

For a more quantitative seismic interpretation, à priori information of the physical properties of mineral and fluid phases occurring at depth has to be taken into account. For instance, Julian et al. (1996) interpreted $V_P/V_S$ anomalies observed in The Geysers, California, in terms of steam pressure, based on a mixing law of fluid and rock mineral properties. A more common way to incorporate fluid properties is through well-known fluid substitution theories, such as those of Gassmann (1951) and Biot (1956a and 1956b), together with estimates of the rock frame mechanics (e.g., Nur and Simmons, 1969; Dvorkin et al., 1999). Husen et al. (2004) processed local earthquake tomography data from the Yellowstone volcanic field, Wyoming, while Vanorio et al. (2005) carried out a comparable study of data from Campi Flegrei, Italy. They concluded from fluid substitution calculations that $V_P/V_S$-ratio anomalies were caused by gaseous pore fluids. De Matteis et al. (2008) acquired tomograms in



the Lardarello-Travale geothermal field in Italy and used the fluid substitution theory to identify steam bearing formations, condensation zones, and over-pressured zones.

Other advanced petrophysical models consider fluid inclusions of specific shape, usually simplified as spheres and ellipsoids, for example those reported by Kuster and Toksöz (1974) or Kachanov et al. (1994). Such a model was applied for seismic interpretation by Tryggvason et al. (2002), who modeled fractured rock as fluid inclusions of ellipsoidal shape, with the fluids having properties either of supercritical water or partial melt. Based on these calculations, they interpreted a low $V_P/V_S$ anomaly in Hengill, Southeast Iceland, as a region containing fractures saturated with supercritical water and excluded the presence of partial melting in the same region. Adelinet et al. (2011) interpreted a $V_P$ and $V_S$ anomaly below the Reykjanes Peninsula in Iceland and delineated a region with over-pressured supercritical fluids by fitting the observed velocities to velocities obtained from an effective medium model. The latter was a function of crack density, crack aspect ratio and liquid-versus-supercritical fluid content.

It is important to note, as highlighted by Mavko et al. (2009), that effective medium models, as traditionally used for rock physics-based seismic interpretation, represent the un-relaxed state (high-frequency limit), in the cases where fluid properties are directly included in the effective medium model. At the other extreme, where fluid saturation in originally dry rock frames is modeled by using fluid substitution techniques, the effective medium describes the rock mechanics in the relaxed state (low frequency limit). Between these high and low frequency limits, seismic wavespeed shows marked dispersion and in addition, strong frequency-dependent seismic attenuation is observed in reservoir rocks, as a result of energy dissipation associated with pore fluid flow triggered by stress-induced pore pressure gradients. This effect is referred to as wave-induced fluid flow, and is caused by various mechanisms, depending on the frequency spectrum of the seismic wave. Velocities at ultrasonic frequencies are affected by global (Darcy) flow due to macroscopic wavelength-scale pressure gradients (Biot, 1956a, b). In the intermediate sonic frequency range, velocities and attenuation are influenced by squirt flow from microscopic compliant cracks into more stiff pores (Winkler, 1985). At low seismic frequencies, seismic properties are affected by localized fluid flow between mesoscopic inhomogeneities of different compressibility (Pride et al., 2004).

All these effects control how pore fluids, depending on their compressibility and viscosity, and the hydro-mechanics of a fractured host rock, leave their footprint on the seismic response of the reservoir, expressed in terms of frequency dependent $V_P$, $V_S$, $Q_S$, and $Q_S$. Thus, consideration of wave-induced fluid flow has a large potential for further improving the rock physics-based seismic interpretation. Moreover, it needs to be recognized that seismic techniques cover a wide range of frequencies, from less than 1 Hz for local earthquake tomography, to more than 100 Hz for active seismic investigations, to the tens of kHz-range for sonic borehole tools, and up to 1MHz for piezo-electric pulse experiments in the laboratory. Thus, it is important to not only model the seismic response of fractured rock at the low- and high-frequency limit, but also at intermediate frequencies.

Different analytical approaches exist to account for velocity dispersion and attenuation due to wave induced fluid flow. For instance Chapman (2003) describe the relaxation of fluid pressure between fluid inclusions, where the compliance of the inclusions is obtained from Eshelby (1957)'s theory. By contrast, Pride et al. (2004) and Gurevich et al. (2009) used Biot (1941)'s theory of poroelasticity, whereas Liu et al. (2009) used the theory of viscoelasticity to consider fluid flow in a double-porosity medium. Such theoretical models are based on some simplifying assumptions such as low fracture density or



small elastic property contrasts, together with idealized geometries of heterogeneities. Motivated by this, numerical modeling approaches, based on the theory of poroelasticity as in the case of Masson and Pride (2007), Rubino et al. (2008), Wenzlau et al. (2010), and Quintal et al. (2011), became popular during the last decade, to complement analytical models.

In this study, we use a numerical modeling technique, which is similar to those proposed by Rubino et al. (2008) and Quintal et al. (2011), to compute the seismic phase velocities and the frequency dependent wave attenuation in fluid saturated fractured reservoirs. The reservoir is embedded in a magmatic-type environment, as it is typical for Iceland. We first define the physical properties of intact rocks based on the results of laboratory experiments reported in the literature. We take into account the diversity of typical rock types, which are shown to exhibit a large variability of hydro-mechanical properties. Then, for the up-scaling to the dimensions of macro-fractures, we study the properties of individual fractures in dependence of the hosting intact rock using a semi-analytical effective medium approach, which is based on Eshelby (1957)'s elastic field theory. Once the parameters, which describe the physics of fractured rock volumes, are defined for ambient confining pressures under which the fractures are considered to be open, we study how each of these parameters depends on lithostatic stress, under which fractures close gradually. After this parameterization study, we finally apply the numerical model to a fractured geothermal reservoir in Iceland, as described in the structural geology literature. We examine how the frequency-dependent seismic properties of a rock containing a fracture network are affected by its saturating fluid, and how the observed fluid effects differ, depending on the hosting lithology and on the effective lithostatic stress.

## 2 Theoretical background

### 2.1 Numerical poroelastic modeling

To study the effects of fluids on the seismic properties of fractured rock, we use a numerical modeling technique which is based on the work of Quintal et al. (2011). It primarily involves Biot's (e.g. 1941) theory of poroelasticity and the principle of conservation of linear momentum;

$$\nabla \cdot \boldsymbol{\sigma} = 0, \tag{1}$$

where $\boldsymbol{\sigma}$ is the stress tensor, whose components in 2-D are related to the corresponding elements of the strain tensor $\boldsymbol{\epsilon}$ by the constitutive law

$$\sigma_{ij} = 2G_{\mathrm{d}}\epsilon_{ij} + \lambda\left(\epsilon_{11} + \epsilon_{22}\right)\delta_{ij} - \alpha P_{\mathrm{pore}}\delta_{ij}. \tag{2}$$

Here $\alpha = 1 - K_{\mathrm{d}}/K_{\mathrm{s}}$ is the Biot-Willis coefficient, $\lambda = K_{\mathrm{d}} - 2/3G_{\mathrm{d}}$ is Lamé's constant, and $\delta_{ij}$ is the Kronecker delta. The quantity $K_{\mathrm{d}}$ is the drained frame bulk modulus, $G_{\mathrm{d}}$ is the drained frame shear modulus, and $K_{\mathrm{s}}$ is the bulk modulus of the solid (grain) phase of the porous rock. The drained state is equivalent to no fluid in the pores. The first two terms on the right hand side of (2) are consistent with Hooke's law of linear elasticity, whereas the additional term $\alpha P_{\mathrm{pore}}\delta_{ij}$ accounts for the stiffening of the rock in response to a pore pressure $P_{\mathrm{pore}}$. Biot (1941) completed his theory by adding the conservation of fluid mass, under the assumption of fluid incompressibility. This requires that the flow rate into or out of an element of rock,



described by Darcy's law for a global flow of liquid in a porous medium, is equal to the temporal change of fluid volume due to the deformation of the rock mass and due to the change of pore pressure. Transforming the mathematical formulation used by Quintal et al. (2011) into the space-frequency domain, the fluid transport equation is given by:

$$
-\frac{k}{\eta_\mathrm{f}}\nabla^2 P_\mathrm{pore} + i\omega\alpha\left(\epsilon_{11} + \epsilon_{22}\right)
$$

$$
+ i\omega\left(\frac{\phi}{K_\mathrm{f}} + \frac{\alpha-\phi}{K_\mathrm{s}}\right)^{-1} P_\mathrm{pore} = 0, \tag{3}
$$

where the imaginary quantity $i$ and the angular frequency $\omega = 2\pi f$ represent the frequency domain-equivalent of the time derivatives. Quantity $k$ is the hydraulic permeability, $\eta_\mathrm{f}$ is the fluid viscosity, $K_\mathrm{f}$ is the fluid incompressibility, and $\phi$ is the effective porosity.

To compute the poroelastic response of the medium, we simultaneously solve Eqs. (1) to (3) for the stress relaxation resulting from an imposed strain, using the COMSOL Multiphysics®finite element solver. In its poroelastic representation on a finite element grid, a fractured rock as observed in nature, containing micro-fractures and macro-fractures of complex shape (Fig. 1a), is defined in a simplified manner (conceptual representation) by a composite of two poroelastic phases – the rock domain and the fracture domain (Fig. 1b). The rock domain represents the parts of the rock which are intact, apart from microscopic cracks which are not discretized individually, and it will be referred to hereafter as the *intact rock*. The fracture domain comprises all the macroscopic fractures, which are in the model individually represented by smooth elliptic structures. We simply refer to them as *fractures* in what follows. As evident from (2) and (3), the hydro-mechanical behavior of each of these two media depends on a set of parameters, which are $K_\mathrm{d}$, $G_\mathrm{d}$, and $K_\mathrm{s}$, $\phi$, and $k$ for the solid phase of intact rock and fractures, and $\eta_\mathrm{f}$ and $K_\mathrm{f}$ for the saturating fluid phase. To distinguish between properties of the two media, we will mark intact rock properties with a hat superscript ('^') and the fracture properties with a tilde superscript ('~') throughout the text.

The model domain has undrained boundaries, meaning that there is no fluid flow across them. To conduct an oscillatory compressibility test, we simulate a vertical normal stress by a displacement disturbance $\Delta u$ in $x_1$-direction to the top boundary, when referring to the coordinate frame in Fig. 2a, and we suppress any displacements in $x_2$-direction at the left and right boundaries, and any displacement in $x_1$-direction at the bottom boundary, as defined in Eq. (A1) in Appendix A. The stress-strain ratio resulting under these conditions yields the complex P-wave modulus, which is for a P-wave propagating towards the $x_1$-direction defined by

$$
M_\mathrm{c}(\omega) = \frac{\langle\sigma_{11}\rangle}{\langle\epsilon 11\rangle}. \tag{4}
$$

For an oscillatory shear test, we apply a displacement $\Delta u$ in $x_2$-direction to the top-boundary, and suppress any displacement in $x_2$-direction at the bottom-boundary, while particles on the left and right boundaries are free to move in both directions $x_1$ and $x_2$, as summarized in Appendix A by Eq. (A2). From the stress-strain relation calculated by this shear-test, we obtain the frequency-dependent complex shear-wave modulus for a S-wave propagating towards the $x_2$-direction from the relation

$$
G_\mathrm{c}(\omega) = \frac{1}{2}\frac{\langle\sigma_{12}\rangle}{\langle\epsilon 12\rangle}. \tag{5}
$$



The angle brackets $\langle \rangle$ in Eqs. (4) and (5) denote the average over the entire modeling domain. Knowing the bulk density of the rock $\rho_{\mathrm{b}}$, seismic phase velocities can be obtained from the complex elastic moduli by (e.g., Casula and Carcione, 1992)

$$V_{\mathrm{P}}(\omega) = \left[ \mathrm{Re} \left( \sqrt{\frac{\rho_{\mathrm{b}}}{M_{\mathrm{c}}(\omega)}} \right) \right]^{-1} \tag{6}$$

and

$$V_{\mathrm{S}}(\omega) = \left[ \mathrm{Re} \left( \sqrt{\frac{\rho_{\mathrm{b}}}{G_{\mathrm{c}}(\omega)}} \right) \right]^{-1}. \tag{7}$$

The attenuation factors are defined as the inverse P- and S-wave quality factors by (e.g., Casula and Carcione, 1992)

$$Q_{\mathrm{P}}^{-1}(\omega) = \frac{\mathrm{Im}\left(M_{\mathrm{c}}(\omega)\right)}{\mathrm{Re}\left(M_{\mathrm{c}}(\omega)\right)} \ \ \text{and} \ \ Q_{\mathrm{S}}^{-1}(\omega) = \frac{\mathrm{Im}\left(G_{\mathrm{c}}(\omega)\right)}{\mathrm{Re}\left(G_{\mathrm{c}}(\omega)\right)}. \tag{8}$$

These key seismic properties resulting from numerical poroelastic modeling have incorporated the dispersive nature of propagation due to the frequency-dependent interplay between the elastic deformation of the fractured rock and the viscous fluid flow in the pores and fractures, as described by Eq. (3). It involves different mechanisms such as localized fluid flow in porous background and squirt-type flow in fractures.

The price for getting such a detailed characterization of seismic properties is that the method requires the determination of a large set of parameters. In this study, we will give a detailed overview of typical values and likely ranges for each of these parameters, while accounting for the large diversity of rock types in magmatic geothermal systems. Furthermore, we will study how these values depend on lithostatic stress. A problematic feature with this approach is that parameterizing individual fractures by a homogeneous poroelastic medium and not by fluid filled cavities is a more conceptual rather than a direct physical representation. In particular, the definition of the fracture stiffness by intrinsic specific elastic moduli $\tilde{K}_{\mathrm{d}}$ and $\tilde{G}_{\mathrm{d}}$ neglects the fact that in reality the elasticity of an open fracture is a complex interplay between the geometry of the void and the elasticity of the surrounding intact rock, which also involves a changing behavior of the intact rock due to the presence of the fracture, as has been described by Eshelby (1957).

## 2.2 Semi-analytical effective medium modeling

To study the properties of individual fractures under dry conditions, as required to determine the dry frame elastic moduli $\tilde{K}_{\mathrm{d}}$ and $\tilde{G}_{\mathrm{d}}$, we use the semi-analytical solution provided by the effective medium theory. The effective elasticity of a composite material, consisting of an isotropic elastic intact rock containing ellipsoidal inclusions which are filled with an isotropic elastic material (or empty as in our case), is calculated with the Mori-Tanaka method (Mori and Tanaka, 1973). An expression for the effective elastic tensor for the case where the ellipsoids are randomly distributed in a plane (holding one axis of the ellipsoids fixed, whereas the other two are randomly oriented as shown in Fig. 2b), is given by Pan and Weng (1995) as

$$\mathbf{C}_{\mathrm{Eff}}^{-1} = \left( \mathbf{I} + c_{\mathrm{f}} \mathbf{A} \right) \mathbf{C}_{\mathrm{m}}^{-1}. \tag{9}$$

Here, $\mathbf{C}_{\mathrm{m}}$ is the elasticity tensor (in Voigt's matrix notation) of the intact rock, $c_{\mathrm{f}}$ the volumetric concentration of inclusions, $\mathbf{I}$ the identity matrix and $\mathbf{A}$ is the eigenstrain concentration tensor, describing the strain under zero stress. The latter quantity is





defined by Pan and Weng (1995) to be

$$\mathbf{A} = -\mathbf{Q}\left(\mathbf{I} + c_{\mathrm{f}}\mathbf{P}\right)^{-1}, \tag{10}$$

where

$$\mathbf{P} = \left\langle (\mathbf{I} - \mathbf{S})\left[(\mathbf{C}_{\mathrm{f}} - \mathbf{C}_{\mathrm{m}})\,\mathbf{S} + \mathbf{C}_{\mathrm{m}}\right]^{-1}\right\rangle (\mathbf{C}_{\mathrm{f}} - \mathbf{C}_{\mathrm{m}}) \tag{11}$$

and

$$\mathbf{Q} = \left\langle \left[(\mathbf{C}_{\mathrm{f}} - \mathbf{C}_{\mathrm{m}})\,\mathbf{S} + \mathbf{C}_{\mathrm{m}}\right]^{-1}\right\rangle (\mathbf{C}_{\mathrm{f}} - \mathbf{C}_{\mathrm{m}}). \tag{12}$$

In (11) and (12), angle brackets $\langle \cdot \rangle$ denote the orientational average of the corresponding tensor, given in Appendix B. Quantity $\mathbf{C}_{\mathrm{f}}$ is the elasticity tensor of the fracture filling-fluid phase and $\mathbf{S}$ is the Eshelby (1957) tensor, whose components for ellipsoidal inclusions can be calculated (e.g., Mura, 1987) from the aspect ratio of the ellipsoids and the elastic properties of the intact rock, i.e. from $\hat{K}_{\mathrm{d}}$ and $\hat{G}_{\mathrm{d}}$. Due to the random orientation of the ellipsoids in the $x_1$-$x_2$ plane, the resulting effective elasticity tensor is transversely isotropic and the velocities of P- and S-waves propagating along the $x_1$-axis are calculated from the corresponding components of the elasticity tensor $\mathbf{C}_{\mathrm{Eff}}$ by

$$V_{\mathrm{P}} = \sqrt{\frac{C_{11}}{\rho_{\mathrm{b}}}} = \sqrt{\frac{C_{22}}{\rho_{\mathrm{b}}}} \ \text{ and } \ V_{\mathrm{S}} = \sqrt{\frac{C_{44}}{\rho_{\mathrm{b}}}} = \sqrt{\frac{C_{55}}{\rho_{\mathrm{b}}}}. \tag{13}$$

The effective medium theory is subject to several limitations in terms of the geometrical representation of fractured rock. The underlying theory is exact only for non-interacting fractures (Kachanov, 1992), and is consistent with the upper and lower Voigt-Reuss bounds only in the case of low volumetric fracture density (Berryman and Berge, 1996). Furthermore, as stated by Kachanov (1992), the assumptions of non-interacting fractures and of small fracture density are not equivalent, since for non-randomly located fractures, the interaction might still be strong even for a dilute fracture density. For these reasons, fractures are considered to be well separated from each other and randomly located in space.

### 2.3 Dry fracture elasticity estimation

Compared with the benefits and drawbacks of the numerical modeling technique, the semi-analytical effective medium theory has complementary cons and pros. The effective medium is limited to non-interactive fractures, while the poroelastic theory implemented on a finite element gird allows modeling the hydro-mechanical interaction of complex fracture networks. On the other hand, the effective medium theory implicitly includes the stiffness of the fractures, depending on the intact rock elasticity and the geometry of the fractures, whereas the numerical technique requires parameterizing individual fractures by a poroelastic medium, where $\tilde{K}_{\mathrm{d}}$ and $\tilde{G}_{\mathrm{d}}$ are treated as fracture intrinsic material properties.

In the parameterization part of this paper, we will combine the two techniques to obtain appropriate values for the dry frame fracture stiffness's $\tilde{K}_{\mathrm{d}}$ and $\tilde{G}_{\mathrm{d}}$ by varying $\tilde{K}_{\mathrm{d}}$ and $\tilde{G}_{\mathrm{d}}$ until the stiffness of the overall fractured rock resulting from the





poroelastic numerical modeling is consistent with that from the effective medium theory. To ensure that the 2-D numerical

fractured rock model satisfies the requirements of the effective medium model, we generate models of randomly located, randomly oriented and well-separated fractures of thin elliptic shape (black lines in Fig. 2a). The volumetric concentration of fractures is below 1%, what is below the limit for the low-fracture density assumption of 10-20% determined by Berryman and Berge (1996). We define an ellipse-shaped fracture-free area around each fracture (dotted ellipses in Fig. 2-a), and to guarantee that the fracture orientation is not biased by neighboring fractures, we successively place fractures within a circular

area (dashed circle in Fig. 2a), which allows rotating the fracture by 360° independently from the orientation of neighboring fractures. As the numerical modeling is in 2-D, we assume the fractures to extend continuously in the out of plane direction over distances that are long compared to the in-plane fracture dimensions. Therefore, the 3-D effective medium model (Fig. 2b) contains fractures with semi-major axis $a_3$ being much longer than semi-minor axes $a_1$ and $a_2$, whereby the solution of the effective medium theory with ellipsoidal inclusions converts to one from a composite containing elliptic cylinders. The aspect

ratio $a_1/a_2$, the fracture density $c_f$, and the intact rock properties are chosen to match those of the numerical model.

When estimating values of the stiffness $\tilde{K}_d$ and $\tilde{G}_d$ of dry fractures, no fluids are involved and the non-interaction condition is satisfied also for the numerical model in terms of fluid flow between fractures. Once appropriate values of $\tilde{K}_d$ and $\tilde{G}_d$ are found, we will extend the complexity of the numerical fractured rock model beyond the capability of the effective medium theory, giving an example of modeling the seismic response of rocks containing fluid saturated interconnected fracture networks. Also

for this case, the volumetric fracture density is still below 1% and, thus, the low fracture density-condition is still fulfilled. Here, uncertainty arises from the fact, that the surrounding material of individual fractures also includes a small fraction of weaker material as fractures intersect. Uncertainties related to this effect can be reduced by using a more comprehensive effective medium theory than the one presented here, such as the self-consistent approach (Mavko et al., 2009, p. 185), which introduces a fractured background medium in an iterative manner.

## 3 Geology

In the present study, we focus on Icelandic geothermal systems. Iceland is a large subaerial part of the worldwide system of mid-ocean ridges. Thus, the crust is to some degree of oceanic type, but anomalously thick with a maximum Moho depth of around 20 to 40 km (Darbyshire et al., 2000). The crustal sequence has been compared with the classical oceanic ophiolite sequence (Foulger et al., 2003; Bjarnason and Schmeling, 2009), but is of larger structural and chemical complexity (Gudmundsson,

1995). In a brief summary, the stratigraphy of the upper few kilometers of the Icelandic crust can be subdivided into four lithological units (Fig. 3). At the shallowest depths, extrusive rocks dominate, forming interlayered sequences of (A) pyroclastic deposits (hyaloclastits, tuffs, scoria, etc.) and (B) lavaflow deposits (dense and vesicular basalts). In lower regions, (C) dyke and sheet intrusions (dolerite) become more and more abundant, which reach down to depths were (D) intra-crustal crystallized magma chambers (gabbro bodies) exist, which are found at depths as shallow as 1-2 km (Gudmundsson, 1995), but typically they occur at greater depth.



The physical properties of these rock types depend to some degree on their chemistry, which is predominantly of basaltic composition but, to a lesser extent, also magmatic rocks crystallized from intermediate and acid magmas exist (Gudmundsson, 1995). Depending on the temperatures and the intensity of fluid circulation through the formations, the chemistry of the rocks is modified by hydrothermal alteration, what additionally increases the variety of minerals, each with potentially different physical properties. But not only the chemistry influences the physical properties of the rocks, also the rock texture has a strong influence. Dense gabbros are different from e.g. vesicular basalts or a highly porous tuff, independently from their chemical compositions. This also results in a large variability of the seismic properties as has been reported, e.g. by Vanorio et al. (2002) for pyroclastic rocks and by Grab et al. (2015) for basalts, dolerites and gabbros. Accordingly, a large variability is expected for the properties of the intact rock in our models, and a large volume of data is required to provide well-grounded estimates. Pyroclastic deposits are a typical feature of on-land volcanism. Lavaflow deposits, dykes and sheets, and magma chambers can also be found at submarine mid-ocean ridges. Therefore we can include the database of the ocean drilling programs for determining their physical properties.

## 4 Model parameterization for ambient lithostatic stress

The poroelastic model of fractured rock consists of two subdomains, intact rock and fractures. Their solid matrix are described by the same type of parameters. For the intact rock these are the dry frame bulk modulus $\hat{K}_\mathrm{d}$, dry frame shear modulus $\hat{G}_\mathrm{d}$, grain bulk modulus $\hat{K}_\mathrm{s}$, dry bulk density $\hat{\rho}_\mathrm{b}$, effective porosity $\hat{\phi}$, and hydraulic permeability $\hat{k}$. The analogous parameters for the fractures are $\tilde{K}_\mathrm{d}$, $\tilde{G}_\mathrm{d}$, $\tilde{K}_\mathrm{s}$, $\tilde{\rho}_\mathrm{b}$, $\tilde{\phi}$, and $\tilde{k}$.

### 4.1 Intact rock properties

We defined intact rock as those parts of the rock embedding the macroscopic fractures. Due to their limited size, rock samples investigated in the laboratory typically are free of such macro fractures, that is why we will refer to laboratory studies to determine intact rock properties. We here present a compilation of published results, which include more than 500 rock samples in total of diverse types from on-land volcanic systems as well as samples included in the database of the Deep Sea Drilling Program and the Ocean Drilling Program.

Figure 4 illustrates in the form of cross plots values for all solid-constituent parameters as they have been reported in the literature. Assigning each rock sample to one of the main lithologies introduced in Section 3, we can study typical physical properties for each of these lithologies. This is depicted in Fig. 4a for the drained bulk modulus, with the boxes indicating the 25th and 75th percentiles, and dots are values outside these percentiles.

Values for the dry frame elastic moduli, $\hat{K}_\mathrm{d}$ and $\hat{G}_\mathrm{d}$ are obtained from velocities $V_\mathrm{P}$ and $V_\mathrm{S}$, measured in the laboratory at ultrasonic frequencies, from the relations

$$\hat{K}_\mathrm{d} = V_\mathrm{P}^2 \hat{\rho} - \frac{4}{3} V_\mathrm{S}^2 \hat{\rho} \;\; \text{and} \;\; \hat{G}_\mathrm{d} = V_\mathrm{S}^2 \hat{\rho}, \tag{14}$$




provided the wavespeeds were measured on dried rock specimens under drained conditions. In Fig. 4b, $\hat{K}_\mathrm{d}$ is plotted against $\hat{G}_\mathrm{d}$, indicated by the open symbols, together with the elastic moduli of saturated rocks shown by the filled symbols. As for

all other parameters, we seek to establish regression relationships using appropriate functions, to get representative values to parameterize the fractured rock models. For the dry frame elastic moduli, the best-fit relationship was found to be:

$$\hat{G}_\mathrm{d} = p_1 \hat{K}_\mathrm{d}^{p_2} + p_3, \tag{15}$$

with $p_1 = 1.4 \times 10^5$, $p_2 = 0.5117$, and $p_3 = -11.5 \times 10^9$.

The grain bulk moduli $\hat{K}_\mathrm{s}$ were estimated using Gassmann's (1951) fluid substitution theory, which uses $\hat{K}_\mathrm{s}$ together with

$\hat{K}_\mathrm{d}$ to predict the bulk modulus of the saturated rock, whereas $\hat{G}_\mathrm{d}$ is assumed to be independent of liquid saturation conditions. There is evidence for the validity of this supposition in the data shown in Fig. 4b, where the bulk moduli of saturated rocks (filled symbols) tend to be higher than those of dry rocks (empty symbols), and no significant increase is observed for the shear moduli.

Applying the fluid substitution theory to all rock samples for which the seismic velocities were measured under dry and

saturated conditions, we investigated what values of $\hat{K}_\mathrm{s}$ are needed to predict the velocities of saturated rocks from those of dry rocks. Whilst velocities of dry rocks are expected to be frequency independent, strong velocity dispersion is often observed for saturated rocks, especially at low confining pressures, where compliant micro cracks are open. To minimize errors due to frequency effects, seismic velocities measured at high confining pressures were used, resulting in values of $\hat{K}_\mathrm{s}$ as shown in Fig. 4c. From the regression analysis, we find

$$\hat{K}_\mathrm{s} = p_4 \exp\left(p_5 \hat{K}_\mathrm{d}\right) + p_6 \exp\left(p_7 \hat{K}_\mathrm{d}\right), \tag{16}$$

with $p_4 = 5.82 \times 10^{10}$, $p_5 = 3.86 \times 10^{-12}$, $p_6 = 8.22 \times 10^{08}$, and $p_7 = 3.99 \times 10^{-11}$. To test the validity of these estimates, we use $\hat{K}_\mathrm{s}$ together with the effective porosity $\hat{\phi}$ obtained from Eq. (18) introduced below, to predict the saturated bulk moduli from the dry bulk moduli by fluid substitution. The resulting saturated bulk moduli are indicated by the dashed line in Fig. 4b, which agrees well with the observed saturated bulk moduli given by the filled symbols, which includes numerous samples not

being used for the $\hat{K}_\mathrm{s}$-estimation.

Most researchers who measured seismic wavespeeds also documented the density and the porosity of the rock samples in their publications. Densities are plotted against $\hat{K}_\mathrm{d}$ in Fig. 4d, and an exponential relationship is indicated, which yields from the regression analysis the best-fit function:

$$\hat{\rho}_\mathrm{b} = p_8 \exp\left(p_9 \hat{K}_\mathrm{d}\right) + p_{10} \exp\left(p_{11} \hat{K}_\mathrm{d}\right), \tag{17}$$

with $p_8 = 2628$, $p_9 = 1.72 \times 10^{-12}$, $p_{10} = -2898$, and $p_{11} = -1.38 \times 10^{-10}$. Values for the effective porosities are shown in Fig. 4e, and the regression analysis yields the best-fit relationship

$$\hat{\phi} = p_{12} \exp\left(p_{13} \hat{K}_\mathrm{d}\right), \tag{18}$$

with $p_{12} = 0.85$, $p_{13} = -1.13 \times 10^{-10}$.





Since only a few authors measured the hydraulic permeability $\hat{k}$ and seismic wavespeeds together, we refer to different publications to estimate values for $\hat{k}$. They are plotted against the bulk density in Fig. 4f and the best fit was obtained with relation

$$\log\left(\hat{k}\right) = p_{14}\exp\left(p_{15}\hat{\rho}_{\mathrm{b}}\right) + p_{16}\exp\left(p_{17}\hat{\rho}_{\mathrm{b}}\right),$$ (19)

with $p_{14} = -11.80$, $p_{15} = 4.26 \times 10^{-05}$, $p_{16} = -3.60 \times 10^{-03}$, and $p_{17} = 2.51 \times 10^{-03}$.

Equations (15) to (19) fully describe the solid frame of the intact rock as a poroelastic medium. As evident from Fig. 4a, it covers a wide range of different lithologies. For the following modeling of the seismic properties of fractured rock, we select parameters for 4 characteristic models, each representing a different lithological unit which we will refer to as lithology A-D. Lithology A represents a typical pyroclastic rock (blue dots in Fig. 4), lithology B a light lava flow deposit (red dots), lithology C a typical dyke or sheet intrusive (yellow dots), and lithology D a relatively dense gabbro body (purple dots). Corresponding parameters for these four models are shown in Table 1.

## 4.2 Fracture properties

At ambient stress, fractures are assumed to be completely open, meaning that fracture walls are not in contact with each other and they can be represented by open ellipses (Fig. 1b). They are considered to be empty, i.e. containing no fault gauge, thus we set the fracture porosity to a high value, $\tilde{\phi} = 90\%$. Furthermore, the mineral composition is assumed to be homogeneous across both the intact rock and the fracture subdomains, with the grain bulk moduli of the two subdomains being identical, $\tilde{K}_{\mathrm{s}} = \hat{K}_{\mathrm{s}}$. This also has the consequence that the mineral density is the same for both subdomains and the dry bulk density of the fracture is defined as $\tilde{\rho}_{\mathrm{b}} = (1 - \tilde{\phi})\rho_{\mathrm{s}}$, where the density of the mineral phase is $\rho_{\mathrm{s}} = 1/(1 - \hat{\phi})\hat{\rho}_{\mathrm{b}}$.

Estimates for the dry frame elastic moduli of fractures, $\tilde{K}_{\mathrm{d}}$ and $\tilde{G}_{\mathrm{d}}$, are obtained by testing what values of $\tilde{K}_{\mathrm{d}}$ and $\tilde{G}_{\mathrm{d}}$ are needed to obtain the same overall fractured rock stiffness's $M_{\mathrm{Num}}$ and $G_{\mathrm{Num}}$ from numerical modeling as the values $M_{\mathrm{Eff}}$ and $G_{\mathrm{Eff}}$ calculated using the effective medium theory. This test is conducted for a fractured rock model with well-separated non-interacting fractures, which allows comparing the results from the numerical modeling with those of the effective medium theory as discussed above in Section 2.3. As an example, $M_{\mathrm{Num}}$ and $G_{\mathrm{Num}}$ calculated for a model with intact rock properties corresponding to lithology B, and for fractures with an aspect ratio $a_1/a_2 = 400$, are shown in Fig. 5a and b by the colored surface for varying values of $\tilde{K}_{\mathrm{d}}$ and $\tilde{G}_{\mathrm{d}}$. The intersection of this surface with the effective medium solution $M_{\mathrm{Eff}}$ and $G_{\mathrm{Eff}}$ is marked with the red lines. There is no solution where both P-wave- and S-wave moduli from the numerical modeling and effective medium theory coincide exactly. Thus, preferential values of $\tilde{K}_{\mathrm{d}}$ and $\tilde{G}_{\mathrm{d}}$ are determined by seeking the minimum in the root mean square deviation, defined by

$$RMS = \sqrt{\frac{\left[\left(\frac{M_{\mathrm{Num}} - M_{\mathrm{Eff}}}{\hat{M}_{\mathrm{d}}}\right)^2 + \left(\frac{G_{\mathrm{Num}} - G_{\mathrm{Eff}}}{\hat{G}_{\mathrm{d}}}\right)^2\right]}{2}}.$$ (20)

From theory, it is expected that the bulk and shear moduli of dry fractures are of similar magnitude (e.g. Lubbe et al., 2008), assuming $\tilde{K}_{\mathrm{d}}/\tilde{G}_{\mathrm{d}} \to 1$. Experimental results, however, indicate a ratio for dry fractures which is in fact small but larger than



1. Pyrak-Nolte et al. (1990) observed a ratio in the range $1.3 \leq \tilde{K}_d/\tilde{G}_d \leq 5$, Lubbe et al. (2008) reported $1.7 \leq \tilde{K}_d/\tilde{G}_d \leq 5$, and Nakagawa (2013) found the ratio to lie in the range $1.7 \leq \tilde{K}_d/\tilde{G}_d \leq 1.9$. Therefore, we use here a small ratio which is larger than one, $\tilde{K}_d/\tilde{G}_d \approx 1.5$, and we find under this constraint a pair of $\tilde{K}_d$ and $\tilde{G}_d$ values which results in a good agreement (low $RMS$ value) between the numerical modeling and effective medium result, shown by the red dot in Fig. 5a, b and c. This procedure is repeated for all lithologies A-D and for aspect ratios varying between 100 and 600, resulting in $\tilde{K}_d$ and $\tilde{G}_d$ values shown in Fig. 5d and e and listed in Table 2.

The hydraulic permeability of open fractures is defined from well-established empirical relations reported in the hydro-mechanical literature. Based on calculations of laminar flow between two parallel walls, the volumetric flux through a fracture was described by the cubic law to scale with the cube of the aperture (Witherspoon et al., 1980; Zimmerman and Bodvarsson, 1996), leading to hydraulic permeability of the fracture defined as (Mavko et al., 2009)

$$\tilde{k} = \frac{e_h^2}{12}. \tag{21}$$

where the fracture aperture (width) is given by the hydraulic aperture $e_h$. This was defined to be the aperture needed to explain the actually observed flow rate through a fracture with rough fracture walls in a parallel plate model. Thus, $e_h$ can be regarded as a parallel-wall equivalent aperture. Based on experimental observations, a formula for calculating $e_h$ was suggested by Barton et al. (1985) to be:

$$e_h = \frac{JRC^{2.5}}{\left(\frac{h}{e_h}\right)^2}[\mu m], \tag{22}$$

where $h$ is the average mechanical aperture of the fracture. $JRC$ is the joint roughness coefficient with $JRC = 2.5$ indicating a very smooth fracture whereas a fracture with $JRC = 20$ is extremely rough (Barton and De Quadros, 1997). In our models, we use $JRC = 15$, assuming relatively rough fractures.

## 5 Model parameterization as a function of lithostatic stress

To study how the solid frame of the intact rock behaves with depth, we analyze their dependence on the effective confining pressure $P'$, which is defined as the difference between the actual confining pressure (or lithostatic stress) and pore pressure $P' = P_{conf} - P_{pore}$. For the individual fractures we consider the simplest case of an effective stress applied normal to the long fracture axis, given as the normal effective stress $\sigma'_n$.

### 5.1 Intact rock properties as a function of confining pressure

In the laboratory, $V_P$ and $V_S$ are usually measured at various confining pressures, to simulate the lithostatic stress conditions as a function of depth. Such datasets allow one to study the change of the bulk modulus and the shear modulus as a function of confining pressure, approximated by the second order Taylor series expansions,

$$\hat{K}_d(P') = \hat{K}_{d,0} + \frac{\partial \hat{K}_d}{\partial P'}P' + \frac{\partial^2 \hat{K}_d}{\partial P'^2}P'^2 \tag{23}$$



and

$$\hat{G}_{d}(P') = \hat{G}_{d,0} + \frac{\partial \hat{G}_{d}}{\partial P'} P' + \frac{\partial^2 \hat{G}_{d}}{\partial P'^2} P'^2, \tag{24}$$

where $\hat{K}_{d,0}$ and $\hat{G}_{d,0}$ are the respective elastic moduli at zero confining pressures. To estimate values of the first and second order derivatives with respect to the effective confining pressure, we use all entries of the literature database, for which both $V_P$ and $V_S$ were measured at varying confining pressures, and where the pore pressure is known in order to calculate the effective confining pressure. These data have been divided into subsets, with rock samples which show $\hat{K}_{d,0} < 20$ GPa (representing lithology A), $15$ GPa $< \hat{K}_{d,0} < 45$ GPa (lithology B), $40$ GPa $< \hat{K}_{d,0} < 70$ GPa (lithology C) and $65$ GPa $> \hat{K}_{d,0}$ (lithology D), and average values for the derivatives with respect to $P'$ have been calculated for each subset. The resulting bulk and shear moduli as a function of $P'$ for the four lithologies A-D, approximated by substituting the resulting derivatives into Eqs. (23) and (24), are shown in different colors in Fig. 6a and b, together with laboratory data indicated in gray. Values for the derivatives are given in Table 3.

Referring to experimental studies, hydraulic permeability of intact rock as a function of confining pressure has been described by a log-log relationship, e.g. by Lee and Farmer (1993),

$$\log\left(\hat{k}(P')\right) = \log\left(\hat{k}_0\right) - b\log\left(P'\right), \tag{25}$$

with ambient pressure permeability $\hat{k}_0$ and the coefficient $b$ indicating the curvature of the function or the slope when plotting $\log(\hat{k})$ versus $\log(P')$.

To determine values of $b$ for the four cases of lithologies A-D, we analyze the datasets which comprise permeability measurements for intact rock cores at varying confining pressures. They are shown in gray in Fig. 6c. For each dataset, a best-fit curve according to Eq. (25) was obtained. Resultant values for coefficient $b$ are plotted against resulting ambient-pressure permeability $\hat{k}_0$ in Fig. 6d. Since $b$ represents the curvature of the $\log(\hat{k})$ versus $P'$-relationship, its magnitude indicates the sensitivity of the permeability to changes in confining pressure. It is interesting to observe that rocks with intermediate permeability are only slightly sensitive to pressure, whereas both low and high permeability rocks show a stronger sensitivity. Values of $b$ chosen to represent lithologies A-D are shown in different colors in Fig. 6d and listed in Table 3. Resulting graphs for $\hat{k}(P')$ are shown color-coded in Fig. 6c.

The change in intact rock porosity resulting from a change of applied effective pressure was described by Jaeger et al. (2007). They presented an expression for the change of porosity at a specific effective pressure $P'_n$ due to an increment of effective pressure $dP'$, which can be written as a function of the dry frame bulk modulus and the grain bulk modulus:

$$d\hat{\phi}(P'_n) = -\left[\left(1 - \hat{\phi}\left(P'_{n-1}\right)\right)\frac{1}{\hat{K}_d\left(P'_{n-1}\right)} - \frac{1}{\hat{K}_s}\right]dP', \tag{26}$$

where the initial porosity and the dry frame bulk modulus are also functions of the effective confining pressure, given at the initial pressure $P'_{n-1} = P'_n - dP'$. Thus, the porosity at a given effective pressure $P'$ can be calculated stepwise using small pressure increments $dP'$ and updating $\hat{\phi}(P'_{n-1})$ and $\hat{K}_d(P'_{n-1})$ at each step.





The grain bulk modulus $\hat{K}_s$ is assumed to be approximately constant at varying confining pressures. The dry bulk density of the intact rock varies according to the porosity variation, $\hat{\rho}_b = (1 - \hat{\phi}(P'))\rho_s$, assuming the density of the mineral phase $\rho_s$ is constant.

## 5.2 Fracture properties as a function of normal stress

The closure of natural unfilled fractures under normal stress was described by Bandis et al. (1983) as a function of specific normal and tangential fracture compliance. These quantities are related to the dry frame bulk and shear moduli by (Mavko et al., 2009),

$$\frac{1}{\tilde{B}_n} = \frac{\tilde{M}_d}{h} = \frac{\tilde{K}_d + \frac{4}{3}\tilde{G}_d}{h} \quad \text{and} \quad \frac{1}{\tilde{B}_t} = \frac{2\tilde{G}_d}{h}, \tag{27}$$

in the cases where $\tilde{B}_n$ and $\tilde{B}_t$ are the compliances of dry fractures.

Referring to experimental observations, Bandis et al. (1983) described the fracture closure under normal stress being of hyperbolic form, becoming asymptotic to a small non-zero residual aperture. Based on their expressions, we calculate the fracture aperture as a function of normal stress by

$$h(\sigma'_n) = h_0 - dh(\sigma'_n) = h_0 - \frac{\sigma'_n}{\sigma'_n + a\tilde{M}_{d,0}} a h_0, \tag{28}$$

where $\tilde{M}_{d,0}$ is the dry frame P-wave modulus of the fracture and $h_0$ is the mechanical aperture, both at ambient normal stress as indicated with the zero subscript. The coefficient $a$ is defined as the maximum fracture closure coefficient, being the factor relating zero stress aperture to the maximum aperture closure at very high stress, $dh_{max} = a h_0$, which we introduced to eliminate the specific compliance in the expressions of Bandis et al. (1983). Stress-dependent apertures resulting from (23) are given in Fig. 7c for lithologies A-D, with the maximum and minimum values in each case which results from the aspect ratio-dependency of $\tilde{M}_{d,0}$ (Table 2).

An expression for the dry frame elastic moduli of the fractures can also be derived from the work of Bandis et al. (1983), leading to:

$$\tilde{M}_d(\sigma'_n) = \frac{\tilde{M}_{d,0}\left(1 - a\frac{\sigma'_n}{\sigma'_n + a\tilde{M}_{d,0}}\right)}{\left(1 - \frac{\sigma'_n}{\sigma'_n + a\tilde{M}_{d,0}}\right)^2}. \tag{29}$$

This equation gives the P-wave modulus as a function of normal stress, ambient-pressure elasticity, and fracture closure coefficient $a$. Thus, $\tilde{M}_{d,0}$ is an intrinsic material property, which we can define without requiring any information about the absolute aperture such as $h_0$ or $h(\sigma'_n)$.

From Eq. (29), the effective bulk and shear moduli of the fracture can be calculated according to the relations:

$$\tilde{K}(\sigma'_n) = \tilde{M}(\sigma'_n)\frac{(1+\nu)}{3(1-\nu)} \tag{30}$$

$$\tilde{G}(\sigma'_n) = \tilde{M}(\sigma'_n)\frac{(1-2\nu)}{2(1-\nu)}, \tag{31}$$





where $\nu$ is the Poisson's ratio, describing the ratio between bulk and shear modulus, $\nu = (3\tilde{K}_\mathrm{d}/\tilde{G}_\mathrm{d} - 2)/(6\tilde{K}_\mathrm{d}/\tilde{G}_\mathrm{d} + 2)$. We can either assume a constant Poisson's ratio $\nu = \nu_0$, as observed at low normal stress, or use a stress dependent Poisson's ratio $\nu = \nu(\sigma'_\mathrm{n})$, in order to take stress-dependent effects into account, such as the higher resistance against shear, when rough fracture walls become interlocked during the closure at increased normal stress.

In the literature, the fracture compliance at varying normal stress has been experimentally investigated in terms of specific compliances $\tilde{B}_\mathrm{n}$ and $\tilde{B}_\mathrm{t}$ and we cannot directly compare it with the outcome of Eq. (29), because of the scaling by the absolute fracture aperture (Eq. 27). Instead, we can compare the stress-dependent fracture compliance normalized by the initial zero stress compliance, where the absolute fracture aperture cancels out:

$$\frac{\tilde{B}_\mathrm{n}(\sigma'_\mathrm{n})}{\tilde{B}_{\mathrm{n},0}} = \frac{\tilde{M}_\mathrm{d}(\sigma'_\mathrm{n})/h(\sigma'_\mathrm{d})}{\tilde{M}_{\mathrm{d},0}/h_0} = \left(1 - \frac{\sigma'_\mathrm{n}}{\sigma'_\mathrm{n} + a\tilde{M}_{\mathrm{d},0}}\right)^2. \tag{32}$$

This is shown in Fig. 7a, where the $\tilde{B}_\mathrm{n}/\tilde{B}_{\mathrm{n},0}$ ratios reported in the literature are displayed in gray. The normalized P-wave compliances for lithologies A-D, calculated according to the right-hand side of Eq. (32) and using $a = 0.75$, are shown in color, where for each lithology two graphs are shown for the maximum and minimum values, depending on the aspect ratio. Using a constant Poisson's ratio, normalized S-wave compliances are identical to the normalized P-wave compliances. They are given in Fig. 7b for Model A-D again color-coded and compared with the literature $\tilde{B}_\mathrm{n}/\tilde{B}_{\mathrm{n},0}$ ratios in gray. Using a fracture closure coefficient $a = 0.75$, we observe a similar behavior of the fracture compliance as reported in the literature, such as those of Nakagawa (2013) with its strong decrease in fracture compliance already at moderate values of $\sigma'_\mathrm{n}$ or those of Lubbe et al. (2008) which only decrease relatively slowly as $\sigma'_\mathrm{n}$ is elevated to 60 MPa (Fig. 7a and b).

At zero $\sigma'_\mathrm{n}$, we considered the fractures being open and we used the cubic law (Eq. (21) for calculating the hydraulic permeability of fractures. At increased $\sigma'_\mathrm{n}$, the fractures start to close and the two fracture walls come locally into contact with each other. Due to these contacts, as stated by Cook (1992), the reduction of permeability with increasing $\sigma'_\mathrm{n}$ is more rapid than the cube of the joint closure and the cubic law is not valid anymore. For this reason, Cook (1992) extended the cubic law, yielding

$$\tilde{k}(\sigma'_\mathrm{n}) = \frac{e_h(\sigma'_\mathrm{n})^2}{12} \cdot$$
$$\frac{\left(1 + \ln\left(e_h(\sigma'_\mathrm{n})/e_{h,0}\right)\right)^3 \left(e_h(\sigma'_\mathrm{n})/e_{h,0}\right)^3}{2 - \left(e_h(\sigma'_\mathrm{n})/e_{h,0}\right)} + \tilde{k}_\mathrm{res}. \tag{33}$$

In Eq. (33), the first additional term leads to permeability reducing faster with increasing $\sigma'_\mathrm{n}$ than the cube of the fracture closure. The last term is the residual permeability $\tilde{k}_\mathrm{res}$, which incorporates the approximately constant permeability at very high $\sigma'_\mathrm{n}$, where all compliant parts of the fractures are closed and fluid flow takes place through the stiffest pores which remain open. The hydraulic permeability resulting from (33) is shown in Fig. 7d for the maximum and minimum cases of the four lithologies A-D and for $\sigma'_\mathrm{n} = 10^{-14} \ \mathrm{m}^2$.

Assuming that the closure of the fracture is entirely compensated by a decrease of fracture void, whereas the area which is occupied by the material comprising the microscopic roughness remains constant, leads to a fracture porosity as a function of



$\sigma'_\mathrm{n}$, given by:

$$\tilde{\phi}(\sigma'_\mathrm{n}) = \frac{h(\sigma'_\mathrm{n}) - (1 - \tilde{\phi}_0)h_0}{h(\sigma'_\mathrm{n})}. \tag{34}$$

The grain bulk modulus $\tilde{K}_\mathrm{s}$ is assumed to be approximately constant at varying confining pressures, as for the intact rock. The dry bulk density of the fractures varies according to the porosity variation, $\tilde{\rho}_\mathrm{b} = (1 - \tilde{\phi}(\sigma'_\mathrm{n}))\rho_\mathrm{s}$, assuming that the density of the mineral phase $\rho_\mathrm{s}$ is constant.

## 10  6   Example - Fractured rock of variable depth and lithology

### 6.1   Model setup

After determining the hydro-mechanical properties of the intact rock and the fractures, the final task is to model the seismic properties of a rock mass containing an interconnected fracture network, which is saturated with a fluid of specific properties. Here we present a synthetic example using a model containing a fracture network which represents a highly fractured

geothermal reservoir. The network geometry is based on the structural geology observations of Gudmundsson et al. (2002), who examined a highly fractured palaeo-geothermal field associated with the Husavik-Flatey fault in northern Iceland. The network is embedded in a host rock consisting of basaltic lava flow piles, meaning that the petrography is similar to the one we presented above as lithology B. The original depth of the system is estimated to be approximately $1.5\ \mathrm{km}$ below the Earth's surface. Today, the overburden has been largely removed by erosion and the fracture network outcrops at the surface, preserved

in the form of mineral filled veins.

Gudmundsson et al. (2002) described in detail the statistics of the network geometry in terms of the spatial fracture frequency, as well as the orientation, the width, and the length-to-width relationships of the fossil fractures. This gives a complete image of the fracture network as it is required to setup our model. This is done using a model generator, which places fractures randomly within the model domain, incorporating the fracture network statistics by weighting functions which are identical to

the observations from the Husavik-Flatey fault. The resulting model is shown in Fig. 8 (a), together with distributions of the fracture orientations (b), apertures (c), and a cross plot of fracture aperture $a_2$ versus fracture length $a_1$ (d). The gray line in Fig. 8d indicates an aspect ratio $a_1/a_2 = 400$, which is the average aspect ratio observed by Gudmundsson et al. (2002).

The variability of fracture aperture and aspect ratios is not only considered in the model geometry but also when assigning the parameters of the poroelastic media representing the fractures. Fractures of larger aperture exhibit higher permeabilities

according to Eq. (21), and fractures of larger $a_1/a_2$ aspect-ratios are stiffer than those with small $a_1/a_2$-ratios as shown in Fig. 5d and e. Corresponding ranges for the (normalized) fracture compliance are plotted against lithostatic stress in Fig. 7a and b, as they were computed from Eq. (29) using the ambient pressure stiffness's listed in Table 2. Ranges for the fracture permeability as a function of lithostatic stress are computed from Eq. (33) and shown by the maximum and minimum curves in Fig. 7d.

The seismic properties for the fractured rock model were computed by using parameters corresponding to the four lithologies A-D, and for effective lithostatic pressures ranging from ambient pressures up to a maximum of $120\ \mathrm{MPa}$. Depending on the





assumed density distribution of the overburden, the effective confining pressure of a geothermal reservoir situated at 3-4 km depth is around 60 MPa. The intact rock and the fractures were saturated with liquid water of constant properties, viz., a fluid of

incompressibility $K_\mathrm{f} = 2$ GPa, and dynamic viscosity $\eta_\mathrm{f} = 0.001$ Pa s. The P-wave and S-wave elastic moduli, $M$ and $G$, and attenuation factors, $1/Q_\mathrm{P}$ and $1/Q_\mathrm{S}$, are computed according to Eqs. (6) to (8), with $M = V_\mathrm{P}^2(\omega)\rho_\mathrm{b,s}$ and $G = V_\mathrm{S}^2(\omega)\rho_\mathrm{b,s}$, and with $\rho_\mathrm{b,s}$ being the bulk density of the saturated fractured rock. The normal and shear stress-strain relationships are obtained by solving Eqs. (1) to (3) with the COMSOL Multiphysics®finite element solver, using the boundary conditions given in Eq. (A1) for the compressibility test, and those in Eq. (A2) for the shear test. Frequencies were varied over a wide spectrum from

$10^{-2}$ Hz to $10^6$ Hz.

## 6.2   Results

Example results for the deduced seismic properties of the fractured rock are shown in Fig. 9, plotted as the P-wave modulus $M$ (a), S-wave modulus $G$ (b), inverse P-wave quality factors $1/Q_\mathrm{P}$ (c), and inverse S-wave quality factors $1/Q_\mathrm{S}$ (d) against the logarithmically scaled frequency. They were computed for lithology B, undergoing a lithostatic pressure of 15 MPa.

Comparing the elastic moduli with the corresponding attenuation graphs, we observe the typical behavior in accordance with Kramers-Kronig dispersion relation (Mavko et al., 2009). At the same frequencies at which $M$ and $G$ are strongly dispersive with distinct inflection points, $1/Q_\mathrm{P}$ and $1/Q_\mathrm{S}$ reach their local maxima. In the example shown here, these frequencies, referred to hereafter as characteristic frequencies $f_c$, are $f_c = 10^{-1}$ Hz and the less prominent one $f_c = 10^4$ Hz for the P-wave properties, and $f_c = 10^{-1}$ Hz and most pronounced $f_c = 10^5$ Hz for the shear wave properties, the latter having secondary

peaks at around $10^1$ Hz and $10^3$ Hz. Norris (1993) linked the characteristic frequency with the diffusion length $l_d$, over which wave-induced fluid pressure diffusion takes place, by the relation

$$2\pi f_c = \frac{D}{l_d^2}, \tag{35}$$

where $D$ is the hydraulic diffusion coefficient, $D = k/\eta_\mathrm{f}(\phi/K_\mathrm{f} + (\alpha - \phi)/K_\mathrm{s})^{-1}M/M_\mathrm{sat}$, and where $M/M_\mathrm{sat}$ is the ratio of the dry frame P-wave modulus and the undrained (saturated) P-wave modulus. The spatial scales of the fluid pressure diffusion

during numerical oscillation tests can be best inferred from snap shots of the Darcy fluxes $\boldsymbol{q}$, which are deduced from the local permeability values $k$ and the pore pressure gradients through the equation:

$$\boldsymbol{q} = \frac{k}{\eta_\mathrm{f}}\nabla P_\mathrm{pore}. \tag{36}$$

Absolute amplitudes of the fluid fluxes $||\boldsymbol{q}|| = \sqrt{q_1^2 + q_2^2}$ (with $q_i$ being the $i$-th component of the flux vector) occurring under compressional oscillations at frequencies of $10^{-1}$ Hz and $10^4$ Hz are shown in Fig. 10a and b, respectively. Absolute fluxes

arising under shear oscillations at frequencies of $10^{-1}$ Hz and $10^5$ Hz are depicted in Fig. 11a and b, respectively.

At the low frequency of $10^{-1}$ Hz, we observe increased fluxes inside all fractures (see zoom-plots in Fig. 10a and 11a), as well as within large areas of the surrounding intact rock. This shows that during one oscillation cycle, the pore fluid flows from the strongly compressed compliant fractures deeply into the stiffer, and less permeable, intact rock. Thus it is a flow between heterogeneities, with the stiffness and permeability values differing by several orders of magnitudes. It takes place





at scales larger than the pore scale but smaller than the wavelength, why this dispersion mechanism is commonly called the mesoscopic flow (MF) mechanism (e.g. Müller et al., 2010). For the example shown here, fluxes are more widespread for the compression experiment than for the shear experiment, what is in agreement with the larger $1/Q_\mathrm{P}$-magnitude compared with

the magnitude of $1/Q_\mathrm{S}$ at $10^{-1}$ Hz, shown in Fig. 9c and d, respectively. Furthermore, the attenuation peak due to MF occurs as a single maximum without minor side peaks. This is because the characteristic frequency of MF is predominantly controlled by the medium with the lower fluid mobility $k/\eta_\mathrm{f}$ (Quintal et al., 2014), which is the intact-rock subdomain here, having a constant permeability throughout the entire model domain of $7 \times 10^{-18}$ m$^2$. Solving Eq. (35) for the diffusion length using the poroelastic parameters of the intact rock and $f_c = 10^{-1}$ Hz, we find $l_d \approx 0.02$ m, which is in good agreement with the width

of regions with increased fluid fluxes in Figs. 10a and 11a.

At increased frequencies, fluid fluxes into the intact rock become less pronounced (Figs. 10b and 11b), because the shorter oscillation cycles limit the pressure relaxation by fluid flow from the fractures into the intact rock with its low permeability. In the intact rock, fluid flow only takes place within direct vicinity of the fractures and is more pronounced at the tips of individual fractures. Thus, as frequency increases, fluid flow concentrates more and more in the highly conductive fractures. This flow is

driven by pressure gradients between different interconnected fractures, which undergo various degrees of compression, either because they are oriented differently relative to the direction of the applied oscillation stress, or because they are of different stiffness. As we identify fluid flow between different fractures at these higher frequencies, the corresponding dispersion mechanism is equivalent to the squirt flow (SF)-type (e.g. Müller et al., 2010), but at a larger spatial scale. The stress for the compressibility test was oriented along the $x_1$-axis, when referring to the coordinate frame in Fig. 8a, and the stress of the shear

experiment was parallel to the $x_2$-axis and being of the dextral form. Therefore, fluid flow dominates in different fractures for the compressibility and for the shear test, which is why there are different numbers of peaks in the $1/Q_\mathrm{P}$- and $1/Q_\mathrm{S}$-plots, and why they are occurring at different frequencies varying between around $10^{-1}$ and $10^5$ Hz. This range can be explained by the fact that the characteristic frequencies linearly scale with fracture permeability according to Eq. (35), which in the case shown here are within the range of $10^{-8}$ and $10^{-11}$ m$^2$. The diffusion length required to explain the observed characteristic

frequencies is $l_d \approx 0.1$ m which is similar to the half-length of most fracture segments between fracture-intersection points in our model (Fig. 8a).

Next, the seismic properties of all lithologies were modeled. The numerical results obtained for lithostatic pressures of 15 MPa are shown in Fig. 12. The red graphs, representing lithology B, are identical with those shown in Fig. 9, whose characteristics were discussed above. Comparing first the absolute magnitudes of the elastic moduli of all four lithologies,

we observe that the fractured rock models A to D become successively stiffer, as a logical consequence of the increased stiffness of all the rock components. The attenuation peak, which we interpreted to be of MF-type, occurs at characteristic frequencies $f_c \leq 10^1$ Hz. Comparing this peak for the four lithologies A-D, we observe a decrease in amplitude, $A_\mathrm{A} > A_\mathrm{B} > A_\mathrm{C} > A_\mathrm{D}$, which can be explained by the intact rock porosities being strongly decreasing in the order $\hat{\phi}_\mathrm{A} > \hat{\phi}_\mathrm{B} > \hat{\phi}_\mathrm{C} > \hat{\phi}_\mathrm{D}$. Higher porosities entail larger amounts of fluid in the saturated rocks and, hence, more energy is consumed by fluid flow leading to higher attenuation. This effect opposes and dominates over the effect of varying stiffness contrasts, which here are of similar magnitudes for the four lithologies, but which would amplify the attenuation when the stiffness contrasts increase.




Regarding the characteristic frequency, it is observed to decrease for the four models, $f_{c,A} > f_{c,B} > f_{c,C} > f_{c,D}$, which is in agreement with Eq. (35) and the fact that the intact rock hydraulic permeability of the four lithologies progressively decreases

5 as $\hat{k}_A > \hat{k}_B > \hat{k}_C > \hat{k}_D$. The opposite effect is observed for the attenuation peaks at the higher frequencies, which can be related with the SF-type mechanism. For the lithologies A-D the peaks occur with increasing amplitudes $A_A < A_B < A_C < A_D$ and at increasing characteristic frequencies $f_{c,A} < f_{c,B} < f_{c,C} < f_{c,D}$. This is because there is less resistance against the fractures closure under lithostatic stress for the softest rock of type A compared to B, C and subsequently D (see Fig. 7c). Therefore, fractures embedded in lithology D remain the most open and retain their original permeability and fluid saturated pore space

10 the most, which is why the SF-attenuation peak is largest and occurs at the highest frequency for lithology D and subsequently lowers for lithology C, B, and A.

 Finally, to study the effect of lithostatic stress on the seismic properties of fractured rock, the elastic moduli and the seismic attenuation are computed for lithology B, on which an effective lithostatic stress is applied ranging from ambient pressures up to 120 MPa. Numerical results are shown in Fig. 13. We observe the elastic moduli $M$ and $G$ to strongly increase with

15 stress, which is predominantly due to the strong stress-dependence of the fracture stiffness (Fig. 7a and b), together with the less dominant stiffening of the matrix (Fig. 6a and b). Comparing the attenuation peaks for varying stress, we observe the MF-peaks occurring at an approximately constant frequency of $10^{-1}$ Hz, which is consistent with the approximately constant permeability of the intact rock, varying by not more than one order of magnitude for a confining pressure ranging from 0 to 120 MPa. The SF-attenuation peak, however, occurs at a characteristic frequency of $f_c > 10^6$ Hz in the case of zero lithostatic

20 stress, and decreases down to $f_c \approx 10^3$ Hz for a lithostatic stress of 30 MPa. For higher lithostatic stress the characteristic frequency decreases even more, down to frequencies which overlap with those of the MF-type dispersion, where the peaks start to overlap indistinguishably. For both types of attenuation mechanism, the amplitudes decrease with increasing lithostatic stress. This is due to the reduced porosity, and, hence, a reduced pore water content, combined with a stiffening of both, intact rock and fractures at elevated lithostatic stress. This gives rise to lower amplitudes of the wave-induced fluid pressure gradients

25 due to the smaller compressibility contrast. As a consequence of these two effects, the amount of fluid flow is reduced and less energy is dissipated, leading to smaller attenuation peaks at increasing lithostatic stress.

## 7 Discussion and Conclusion

In fluid-saturated fractured rock, viscoelastic interaction between the intact rock, the fractures, and the saturating pore fluid causes velocity dispersion and seismic wave attenuation. The underlying mechanisms have been studied in the past by various

30 researchers, as summarized by Müller et al. (2010), and there is a broad consensus about how the degree of seismic wave attenuation and the characteristic frequency at which it occurs depends on the hydro-mechanical properties of the materials constituting the rock. Petrophysical models which consider such viscous fluid flow are able to link seismic quantities which are measured in geothermal exploration campaigns with the hydrological properties. The reason why these models have not been used routinely to date in seismic interpretation is to a large extent because they depend on many input parameters, some of which are difficult to quantify.





We have determined the input parameters for magmatic geothermal systems, as required in numerical oscillation tests, and wide ranges were observed for most properties when considering the high diversity of magmatic rock types. Most of the

input parameters also depend on lithostatic stress, which is why we provided a compilation of functions to calculate the input parameters for varying effective stress. Using these parameters, we computed the seismic properties of rock volumes containing an interconnected fracture network saturated with liquid water. Results from the numerical modeling demonstrate how seismic velocities and attenuation factors strongly depend on the lithology. This was already established for P- and S-wave velocities in our earlier experimental study (Grab et al., 2015). Here, this is ground-truthed by a large database extracted from the literature,

which shows that the seismic velocity structure of magmatic geothermal systems primarily reflects the subsurface lithology. The effects of reservoir permeability and fluid content are only minor. Interpreting seismic data in terms of hydrological target parameters against the contrast of this background heterogeneity can be achieved by studying the seismic attenuation, i.e., the decrease of seismic amplitudes with increasing distance of travel, and if available, by studying the velocity dispersion, viz., seismic velocity differences at varying frequencies.

Our modeling results show how the magnitudes of seismic attenuation and its dispersion are associated with stiffness contrasts and porosity. Large attenuation peaks were found for a rock volume containing a network of open fractures, which decreased considerably when subjecting the fracture network to elevated lithostatic pressures forcing the fractures to close. The characteristic frequency, at which the attenuation reaches its peak, is linked with the fluid mobility, which is a measure of hydraulic permeability and fluid viscosity. At low seismic frequencies, the attenuation is observed to be controlled by meso-

scopic fluid flow from fractures into the surrounding porous intact rock, with the characteristic frequency linearly scaling with intact rock permeability and fluid viscosity. At sonic up to ultrasonic frequencies, attenuation is associated with squirt flow between interconnected mesoscopic fractures which are compressed to differing degrees during normal and shear oscillations. Here, the characteristic frequency linearly scales with fracture permeability and fluid viscosity.

The spread in the observed critical frequencies illustrates that fluid effects in fractured rock can be detected with various

seismic techniques (passive, active, sonic, etc.) or in the ideal case by the combined use of different seismic techniques to cover a broader frequency spectrum. On the other hand, there seems to be no general rule that governs the frequencies at which the conditions are given to assume either the relaxed state (low frequency limit) or unrelaxed state (high frequency limit), beyond which traditional rock-physics concepts are strictly valid. Thus, concepts which incorporate wave-induced fluid flow, like the one we presented in our study, can help improve the quantitative interpretation of all kinds of seismic data.

In the scope of this study, we modeled the influence of wave-induced fluid flow on seismic properties for the wide range of rock properties, but kept the fluid properties constant at those of liquid water at ambient pressure and temperature. To fully exploit the potential of the modeling technique we presented, it can incorporate changes in fluid incompressibility and viscosity as they vary under phase transitions from liquid to the boiling state and ultimately to the vapor phase. This has interesting application possibilities, as for instance the interpretation of changes in the seismic response measured by time-lapse seismic experiments conducted to monitor changes in the fluid phase during reservoir operation. Saturating the fractured rock with boiling fluid also raises the question of how rock properties vary with elevated temperatures. Our modeling technique is valid only for brittle rocks, but pronounced effects can already be expected at temperatures below the brittle-ductile transition.



Experimental investigations on the stiffness and permeability of intact rock and fractures at elevated temperatures are rare. In general, it is known that an increase in temperature results in softening of the intact rock. Thus, increasing the temperature may cause similar behavior to moving from a stiff lithology to a more compliant one as examined in this study.

**Appendix A: Boundary conditions**

The boundaries $\Gamma$ of the model domain $\Omega$ consist of undrained boundaries. To conduct an oscillatory compressibility test, we
simulate a normal stress by a displacement disturbance $\Delta u$ in the $x_1$-direction to the top boundary $\Gamma^{\mathrm{T}}$, when referring to the coordinate frame in Fig. 2a, and we suppress any displacements in the $x_2$-direction at the left $\Gamma^{\mathrm{L}}$ and right $\Gamma^{\mathrm{R}}$ boundaries, and any displacement towards the $x_1$-direction at the bottom boundary $\Gamma^{\mathrm{B}}$, i.e. rigid boundaries at right, left, and bottom, given by

$$u_1 = \Delta u, (x_1, x_2) \in \Gamma^{\mathrm{T}}$$
$$u_2 = 0, (x_1, x_2) \in \Gamma^{\mathrm{R}} \cup \Gamma^{\mathrm{L}}$$
$$u_1 = 0, (x_1, x_2) \in \Gamma^{\mathrm{B}}. \tag{A1}$$

Accordingly, we apply a displacement in $x_2$-direction to $\Gamma^{\mathrm{T}}$ for the oscillatory shear test, and suppress any displacement towards the $x_2$-direction at $\Gamma^{\mathrm{B}}$,

$$u_1 = 0, (x_1, x_2) \in \Gamma^{\mathrm{B}}$$
$$u_2 = \Delta u, (x_1, x_2) \in \Gamma^{\mathrm{T}}. \tag{A2}$$

Meanwhile, particles on $\Gamma^{\mathrm{T}}$ and $\Gamma^{\mathrm{R}}$ are free to move into both directions $x_1$ and $x_2$.

**Appendix B: Orientational average**

The orientational average of a fourth-order tensor is

$$\langle \mathbf{X} \rangle = \frac{1}{\pi} \int_0^{\pi} \mathbf{X}(\theta) \mathrm{d}\theta. \tag{B1}$$

5  where the rotation around the third principal axis $x_3$ by the angle $\theta$ is obtained by applying the transformation law

$$X_{ijkl}(\theta) = \sum_{p=1}^{3} \sum_{q=1}^{3} \sum_{r=1}^{3} \sum_{s=1}^{3} R_{ip} R_{jq} R_{kr} R_{ls} X_{pqrs}. \tag{B2}$$

with $R$ being the corresponding entry of the rotation matrix

$$\mathbf{R} = \begin{bmatrix} \cos(\theta) & -\sin(\theta) & 0 \\ \sin(\theta) & \cos(\theta) & 0 \\ 0 & 0 & 1 \end{bmatrix}. \tag{B3}$$





This leads to the following expressions for the averaged elasticity tensor $\langle \mathbf{X} \rangle$, given in Voigt's matrix notation:

$$\langle X_{11} \rangle = \langle X_{22} \rangle = \frac{1}{8}(3X_{11} + 3X_{22} + X_{12} + X_{21} + nX_{66})$$

$$\langle X_{33} \rangle = X_{33}$$

$$\langle X_{12} \rangle = \langle X_{21} \rangle = \frac{1}{8}(X_{11} + X_{22} + 3X_{12} + 3X_{21} - nX_{66})$$

$$\langle X_{13} \rangle = \langle X_{23} \rangle = \frac{1}{2}(X_{13} + X_{23})$$

$$\langle X_{31} \rangle = \langle X_{32} \rangle = \frac{1}{2}(X_{31} + X_{32})$$

$$\langle X_{44} \rangle = \langle X_{55} \rangle = \frac{n}{2}(X_{44} + X_{55})$$

$$\langle X_{66} \rangle = \frac{n}{8}(X_{11} + X_{22} - X_{12} - X_{21} + nX_{66}), \tag{B4}$$

where the factor $n$ depends on the definition of the shear components $X_{44}$, $X_{55}$, and $X_{66}$, when transforming the fourth rank elasticity tensor into Voigt's matrix notation.

*Acknowledgements.* The authors would like to thank Céline Mallet from the Institute of Earth Science at the University of Lausanne, Lorenz Grämiger from the Geological Institute at ETH Zürich, and Benny Löffel from the Department of Mathematics at ETH Zürich for their inspiring discussions. Special thanks also go to Thomas Driesner from the Institute for Geochemistry and Petrology at ETH Zürich for his

constructive leadership of the Sinergia project COTHERM2, which builds the overarching framework for this study. It is funded by the Swiss National Science Foundation, grant number CRSII2_160757.



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



**Table 1.** Intact rock properties for the four lithologies A, B, C, and D. Elastic moduli $\hat{K}_d$, $\hat{G}_d$, and $\hat{K}_s$ are given in [GPa], porosity $\hat{\phi}$ in [%], permeability $\hat{k}$ in [m$^2$], and bulk density $\hat{\rho}_b$ in [kg m$^{-3}$].

|  | A | B | C | D |
|---|---|---|---|---|
| $\hat{K}_d$ | 10 | 30 | 55 | 80 |
| $\hat{G}_d$ | 7 | 21 | 32 | 42 |
| $\hat{K}_s$ | 62 | 68 | 79 | 99 |
| $\hat{\phi}$ | 27.46 | 2.87 | 0.17 | 0.01 |
| $\hat{k}$ | $5 \times 10^{-14}$ | $3 \times 10^{-17}$ | $4 \times 10^{-19}$ | $4 \times 10^{-21}$ |
| $\hat{\rho}_b$ | 1945 | 2721 | 2887 | 3016 |



**Table 2.** Intrinsic bulk and shear moduli for the fractures for lithologies A-D and for variable aspect ratios, given with units [GPa].

|  | A | B | C | D |
|---|---|---|---|---|
| $\tilde{K}_{\mathrm{d}}(a_1/a_2 = 600)$ | 0.0028 | 0.0093 | 0.0128 | 0.0178 |
| $\tilde{G}_{\mathrm{d}}(a_1/a_2 = 600)$ | 0.0017 | 0.0059 | 0.0080 | 0.0110 |
| $\tilde{K}_{\mathrm{d}}(a_1/a_2 = 400)$ | 0.0036 | 0.0109 | 0.0162 | 0.0220 |
| $\tilde{G}_{\mathrm{d}}(a_1/a_2 = 400)$ | 0.0022 | 0.0069 | 0.0102 | 0.0135 |
| $\tilde{K}_{\mathrm{d}}(a_1/a_2 = 200)$ | 0.0058 | 0.0162 | 0.0267 | 0.0346 |
| $\tilde{G}_{\mathrm{d}}(a_1/a_2 = 200)$ | 0.0037 | 0.0101 | 0.0168 | 0.0212 |
| $\tilde{K}_{\mathrm{d}}(a_1/a_2 = 100)$ | 0.0096 | 0.0290 | 0.0483 | 0.0606 |
| $\tilde{G}_{\mathrm{d}}(a_1/a_2 = 100)$ | 0.0061 | 0.0183 | 0.0304 | 0.0374 |





**Table 3.** Derivatives for the dry frame elastic moduli with respect to effective confining pressure (calculated for elastic moduli in GPa and pressure in MPa), and $b$-values representing the slope of the log-log permeability-pressure relationship.

| | A | B | C | D |
|---|---|---|---|---|
| $\frac{\partial \hat{K}_\mathrm{d}}{\partial P'}$ | 0.107 | 0.116 | 0.059 | 0.044 |
| $\frac{\partial^2 \hat{K}_\mathrm{d}}{\partial P'^2}$ | -0.00056 | -0.00026 | -0.00014 | -0.00010 |
| $\frac{\partial \hat{G}_\mathrm{d}}{\partial P'}$ | 0.063 | 0.057 | 0.036 | 0.047 |
| $\frac{\partial^2 \hat{G}_\mathrm{d}}{\partial P'^2}$ | -0.00021 | -0.00020 | -0.00011 | -0.00009 |
| $b$ | 1.5 | 0.5 | 0.25 | 2.0 |



Natural fractured rock     Poroelastic medium representation     Effective medium representation

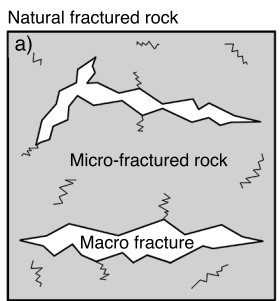 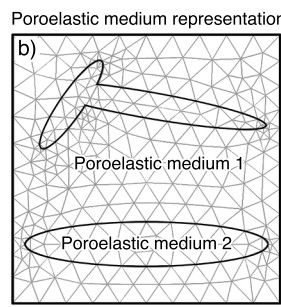 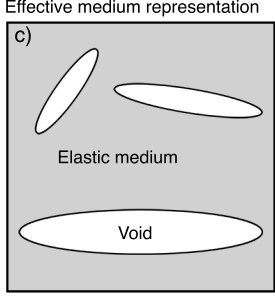

**Figure 1.** Comparison of natural rock with its conceptual representations. a) Natural fractured rock with microscopic and macroscopic fractures of complex shape. b) Poroelastic medium representation, where intersecting macro-fractures and the background medium are both parameterized as isotropic poroelastic media on a finite element grid. c) effective medium representation, consisting of well-separated macroscopic elliptic voids embedded into an isotropic elastic medium.





a) Numerical modeling in 2-D

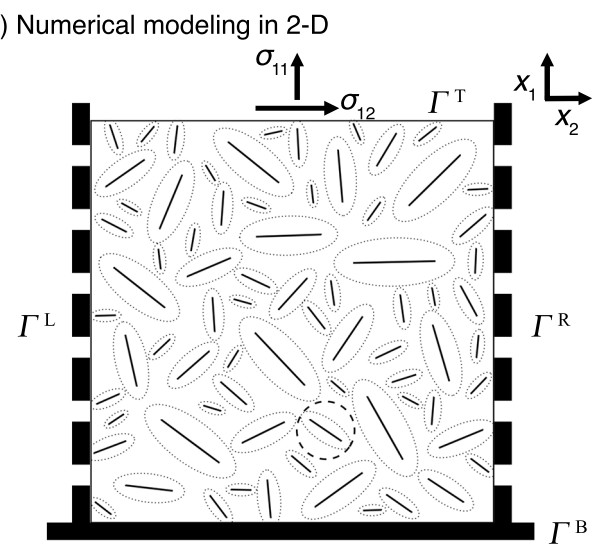

b) Effective medium theory in 3-D

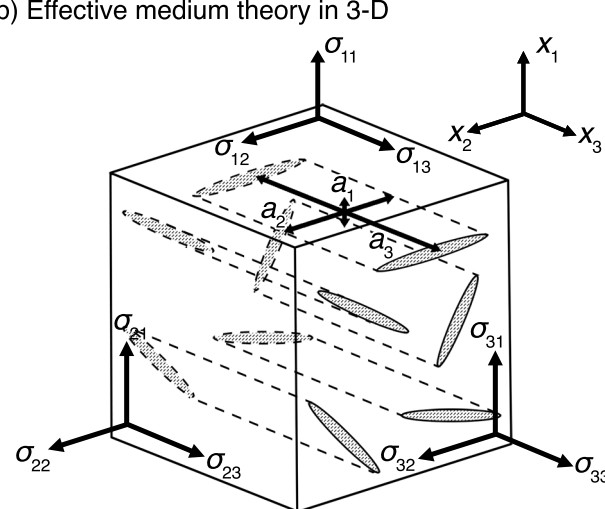

**Figure 2.** a) 2-D numerical modeling scheme for a rock containing randomly oriented, well-separated fractures, to which a normal or shear stress is applied at the top boundary. b) Scheme for 3-D effective medium modeling for a rock containing ellipsoidal fractures, randomly oriented in the $x_1$-$x_2$ plane. Consistent with the numerical model, the applied normal stress $\sigma_{11}$ and shear stress $\sigma_{12}$ is orthogonal to the long ellipsoid axis $a_3$, whose orientation is held constant.



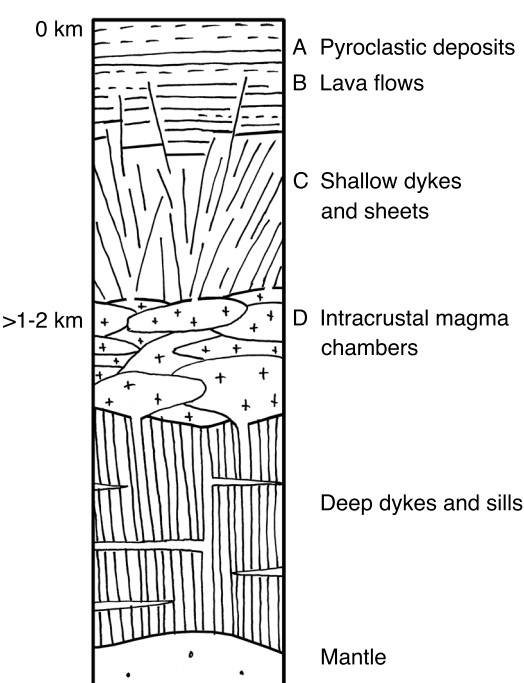

**Figure 3.** Typical stratigraphy of the Icelandic crust. Lithologies A-D are characterized in our study as potential host rocks for geothermal reservoirs.





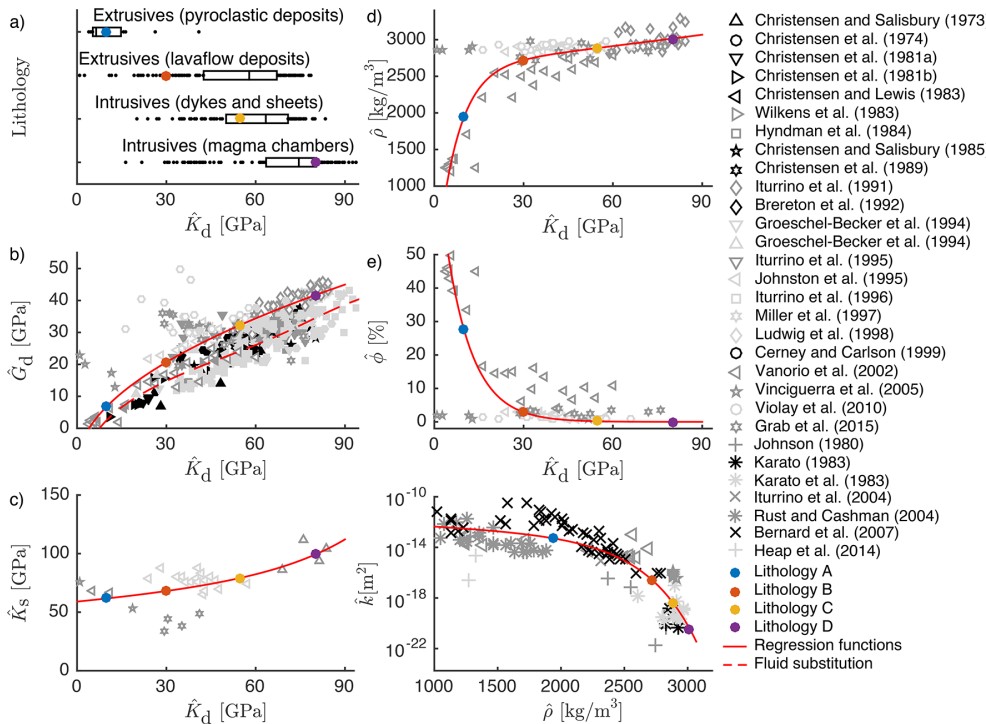

**Figure 4.** Lithological classification (a), physical properties (b-e) of dry (open symbols) and saturated (filled symbols) rocks versus dry frame bulk modulus (and versus saturated frame bulk modulus respectively for saturated samples), and hydraulic permeability versus bulk density (f), as reported in the literature for low confining pressures. Bulk moduli distributions for the lithologies are given as boxes indicating the 25th and 75th percentiles with outliers indicated by the dots. Red lines are best-fit functions obtained from regression analysis, using dry samples only. The dashed line represents the result from the fluid substitution analysis. Bold blue, orange, yellow and purple dots are the values used to parameterize models for lithologies A, B, C, and D.




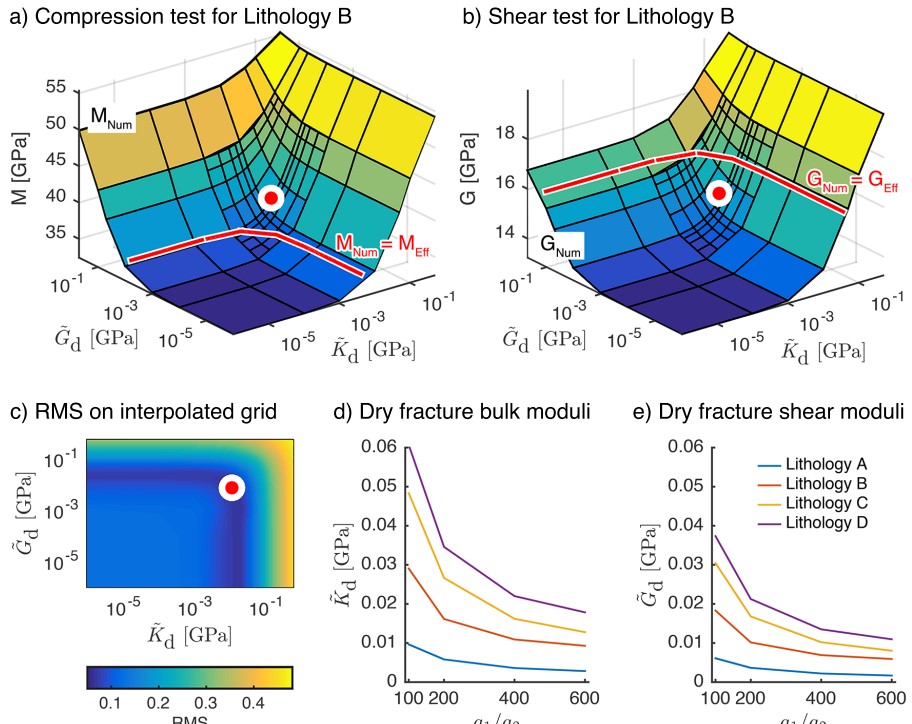

**Figure 5.** Comparison of the fractured rock P-wave modulus (a) and shear wave modulus (b) resulting from numerical modeling for varying fracture elastic moduli (colored surface) with the solution resulting form the effective medium theory (red lines) shown by way of example for lithology B with fracture aspect ratios $a_1/a_2 = 400$. (c) Resulting $RMS$ deviations between the numerical modeling results and the effective medium solution with the minimum (under the assumption of $\tilde{K}_d/\tilde{G}_d \approx 1.5$) indicated by the red dot. (d) and (e): $\tilde{K}_d$ and $\tilde{G}_d$ for all lithologies A, B, C, and D and for the dry frame bulk- and shear moduli, respectively.



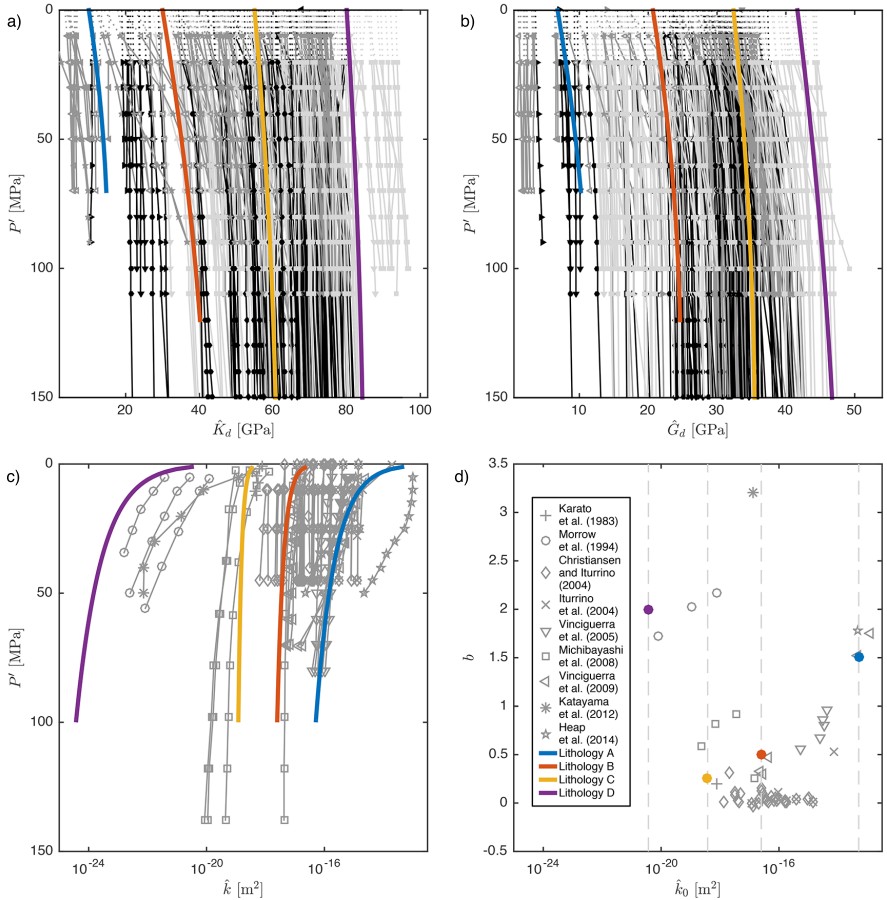

**Figure 6.** Dry frame elastic bulk modulus (a) and shear modulus (b) versus effective confining pressure. Gray and black curves are data reported in the experimental literature (see references in Figure 4), colored curves represent average trends taken for lithologies A-D. (c) Hydraulic permeability versus effective confining pressure and (d) values for the coefficient $b$ versus the ambient confining pressure-permeability. Gray graphs are the permeability data reported in the literature (legend in subplot d) and colored curves and dots indicate values used to parameterize lithology A-D.




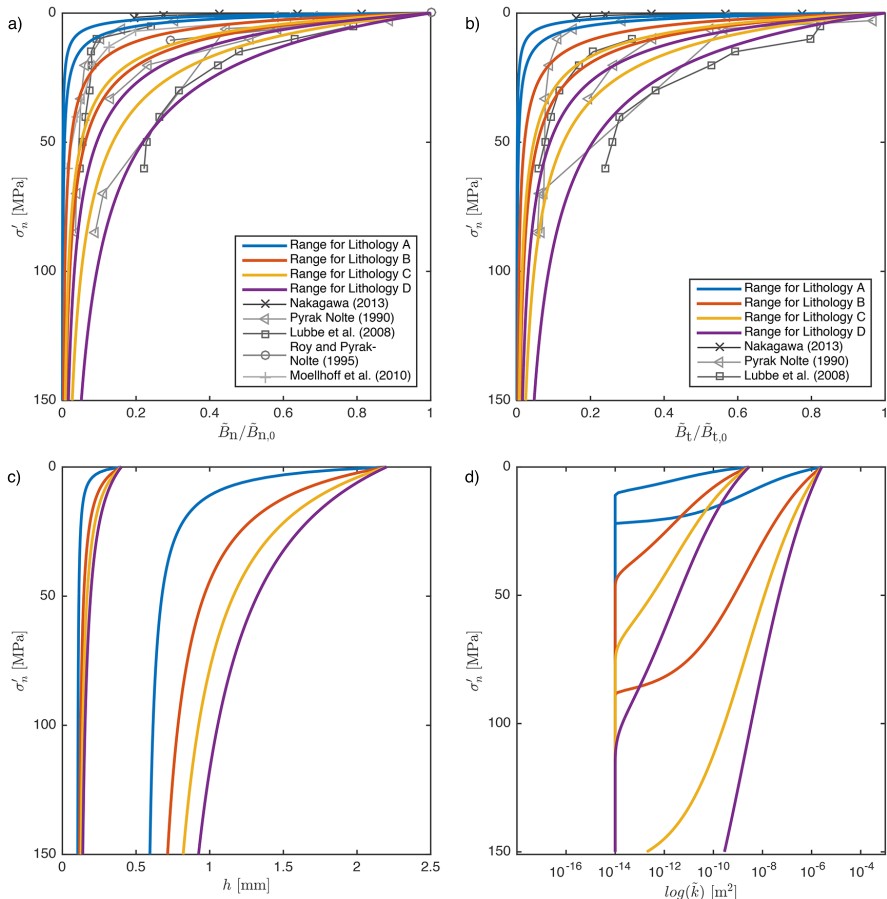

**Figure 7.** Normalized normal (a) and tangential (b) compliances, fracture aperture (c) and hydraulic permeability (d) versus effective normal stress. Gray graphs in (a) and (b) are the experimental observations reported in the literature. Colored graphs are the analytical calculations for lithologies A-D, where for each lithology the maximum and minimum values are given, since all properties vary depending on the fracture aspect ratio or fracture aperture.





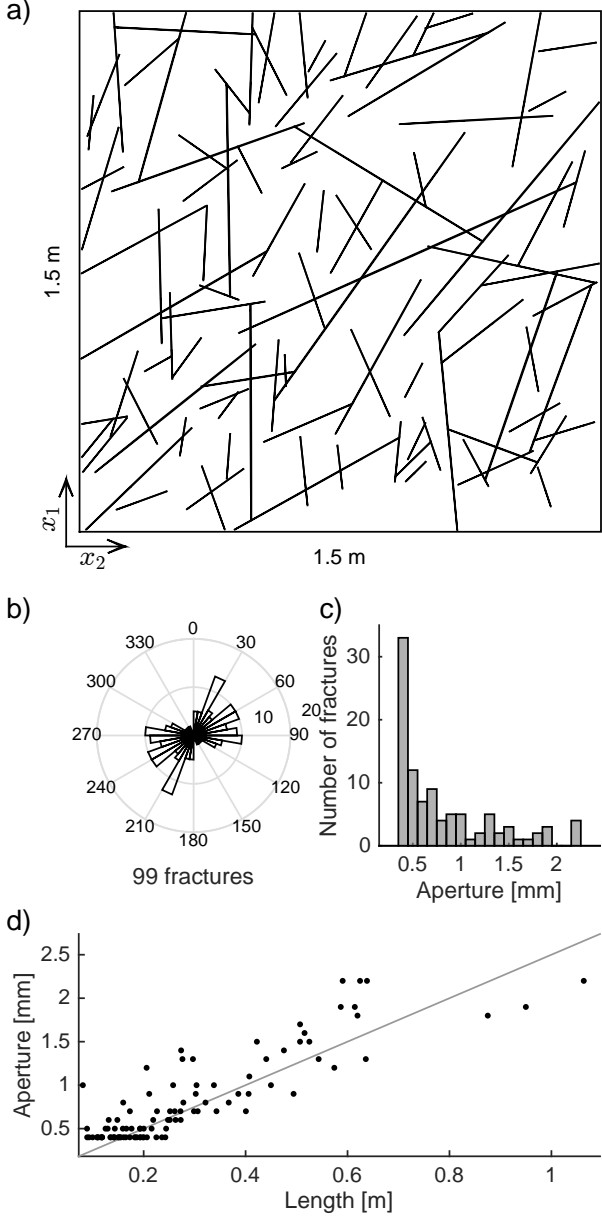

**Figure 8.** Geometry of the fractured rock example (a), and the statistical distribution of the orientations (b), apertures (c), and aperture-length cross plot with the gray line representing values for an aspect ratio of $a_1/a_2 = 400$ (d). Apertures are given as they were defined for zero lithostatic stress.





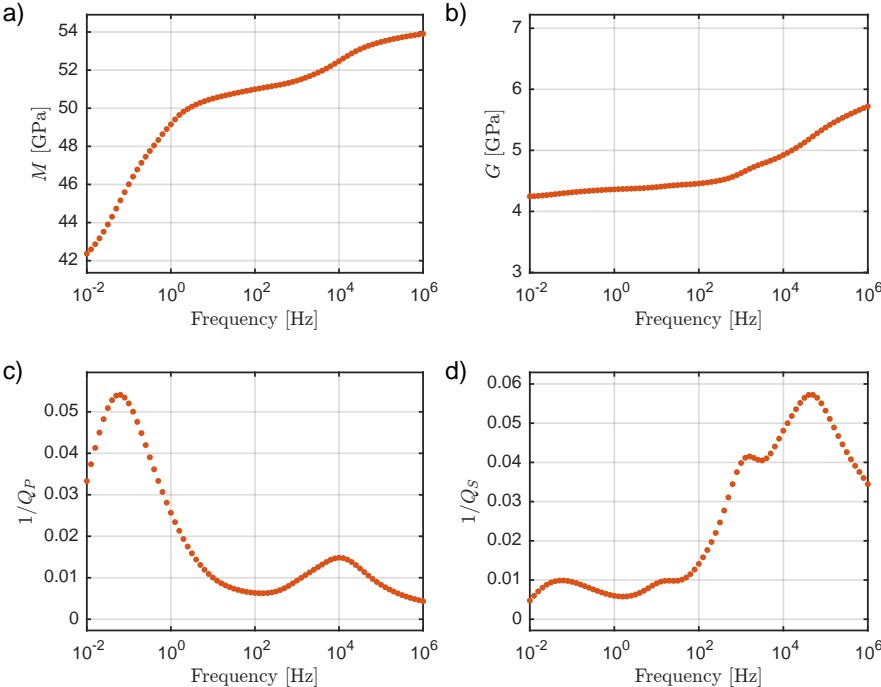

**Figure 9.** P-wave modulus (a), S-wave modulus (b), inverse P-wave quality factor (c) and inverse S-wave quality factor (d), modeled for a fractured rock with the geometry shown in Figure 8, and fracture- and intact rock-properties corresponding to those of lithology B at a lithostatic stress of 15 MPa.





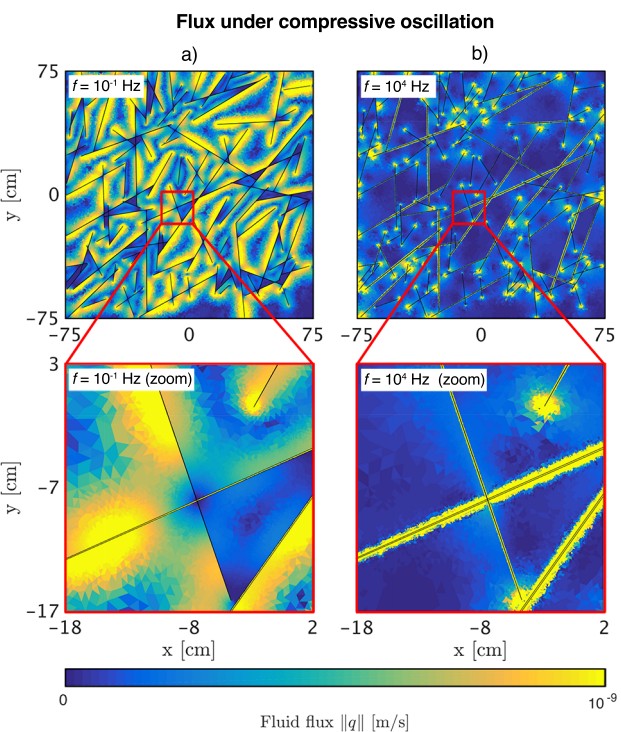

**Figure 10.** Amplitudes of pore fluid fluxes occurring under oscillatory compression at $10^{-1}$ Hz (a) and $10^4$ Hz (b) for the fractured rock properties corresponding to lithology B at a lithostatic stress of 15 MPa (seismic properties for the same case are shown in Figure 9a and c). Subplots at the bottom are enlarged views of specific regions in the top-plots, marked with the red boxes.





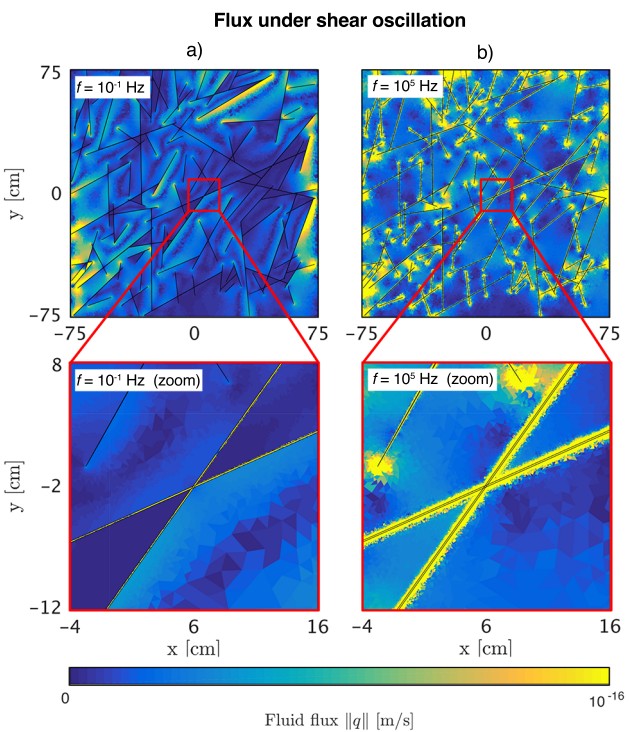

**Figure 11.** Amplitudes of pore fluid fluxes occurring under oscillatory shear at $10^{-1}$ Hz (a) and $10^5$ Hz (b), for the fractured rock properties corresponding to lithology B at a lithostatic stress of 15 MPa (seismic properties for the same case are shown in Figure 9b and d). Subplots at the bottom are enlarged views of specific regions in the top-plots, marked with the red boxes.





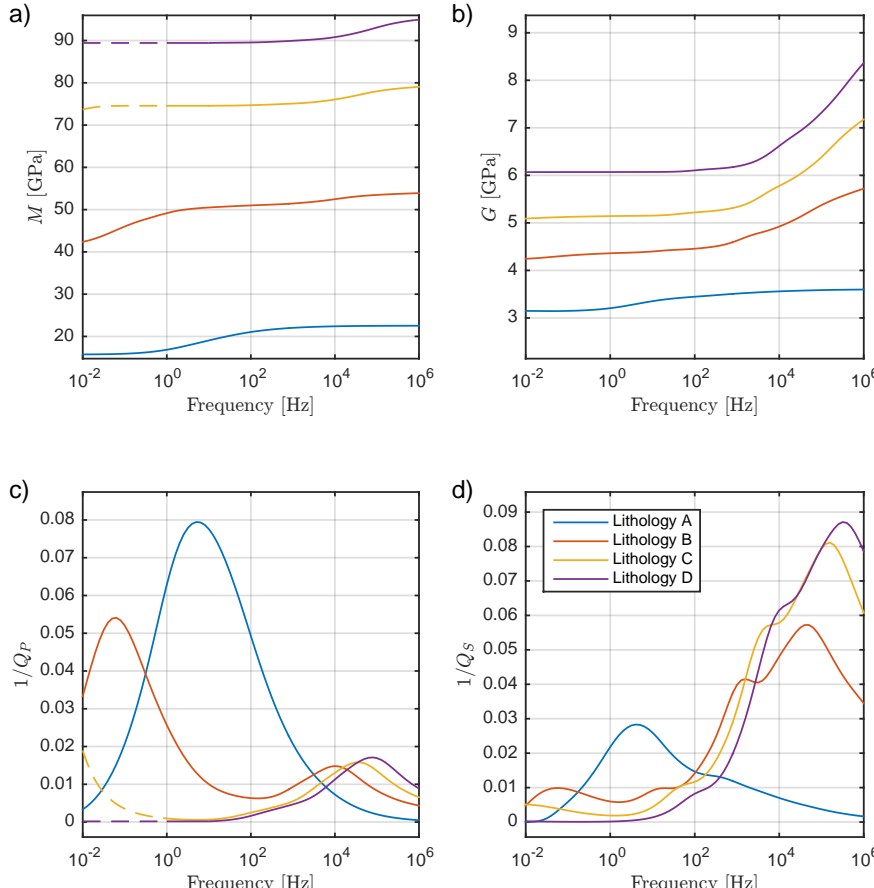

**Figure 12.** P-wave modulus (a), S-wave modulus (b), inverse P-wave quality factor (c) and inverse S-wave quality factor (d) for the four lithology cases A-D, all modeled for a lithostatic stress of 15 MPa. The red curves are identical to those in Figure 9. The dashed parts of the graphs indicate the low frequency ranges, at which the numerical solution breaks down due to the very low permeability and porosity of the corresponding lithologies.





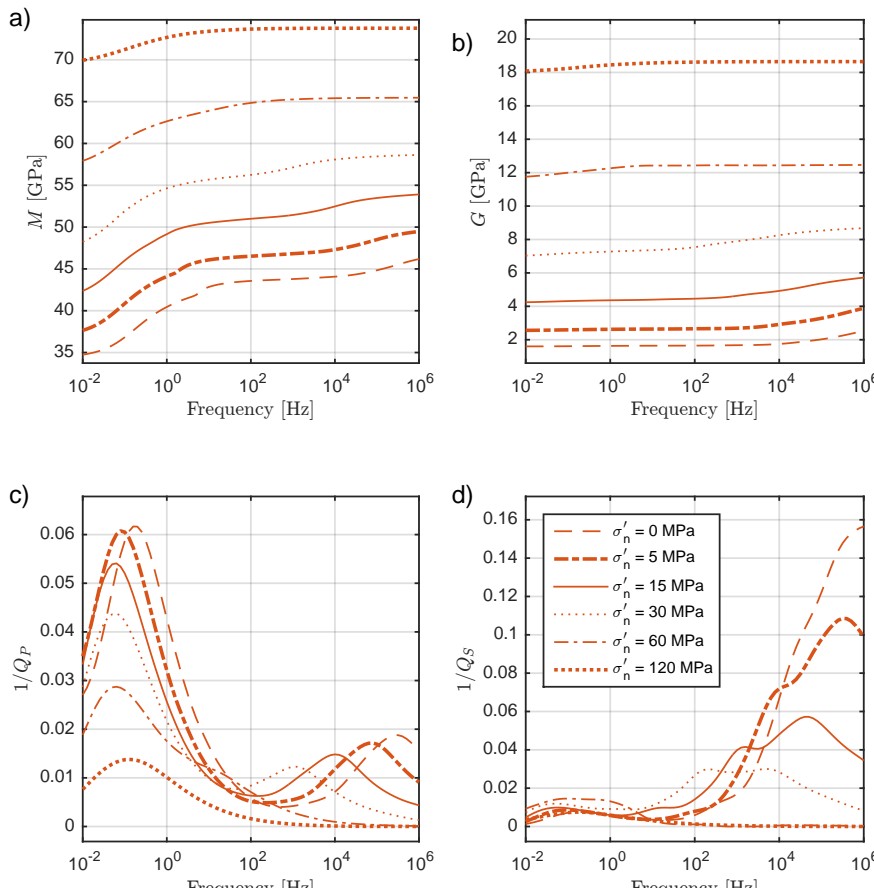

**Figure 13.** P-wave modulus (a), S-wave modulus (b), inverse P-wave quality factor (c) and inverse S-wave quality factor (d) for varying lithostatic stresses, modeled for a fractured rock with a geometry as shown in Figure 8 and rock properties corresponding to those of Model B. The solid curves are identical to those in Figure 9.