# Peer review of "Numerical Modeling of Fluid Effects on Seismic Properties of Fractured Magmatic Geothermal Reservoirs"

_Solid Earth, 2016_

## Referee Comment (RC1) · Anonymous Referee #1 · 16 Dec 2016

The manuscript addresses two main topics. In the first major part rock properties of 4 lithological units of magmatic rocks are derived which are needed for the numerical modeling. Literature data were used to estimate realistic rock properties for magmatic rocks. The second major part is the investigation of the fluid effects on seismic properties (P- and S-velocity and P- and S-attenuation) of fractured rocks by numerical modeling. The numerical modeling covers a wide frequency range from low (passive seismological) over intermediate (active seismological) to high frequencies (laboratory experiments) and can be a link between divergent results of different methods.

I read the manuscript with great interest. I like the results of the numerical modeling showing the frequency dependent effects on the seismic properties of the fractured

rocks and their discussion. I have some comments on the estimation of the rock properties of the 4 lithological units. I suggest to use the classification in the literature and not apply a new classification based on the bulk modulus. I would also derive relations for the rock properties of each lithological unit separately.

Comments:
a.) Classification of lithological units
4 lithological units are introduced: A - pyroclastic deposits, B - lava flow, C - dike sheet intrusions and D - magma chambers.
Figure 4a shows that the data of magmatic rocks found in the literature was classified into lithological units. It would be possible to used this classification. Instead the authors decided to use a classification which is only based on the bulk modulus $K_d$ (page 13, line 8ff) . The 4 new subsets are also labelled as A, B, C and D.

As figure 4a shows that there are lava flow deposits which are now classified as C instead of B. So different types of rocks which may have different rock properties due to their texture (as mentioned) are mixed in the new subsets.

Why was the reclassification performed?
This may lead to misinformation and misinterpretation.
For example subset B is said to represent the lithological unit B - lava flow. (page 13, line 9 and page 16, line 17ff). The properties for subset B is later derived for a bulk modulus $K_d$ of 30 GPa. As can be seen in Figure 4a a rock with a bulk modulus $K_d$ of 30 GPa is an outlier in the lithological unit B.
So the properties of subset B (table 1, 2 and 3) are not(!) representative for lava flow deposits.

An alternative could be to say that the 4 subset are not representing lithological units all, but are arbitrary chosen points within the $K_d$-range at 10, 30, 55 and 80 GPa.

[Figure]

b.) Relations of rock properties (eq. 15, 16, ...)
On Page 9, line 8ff the authors state that the rock properties depends not only on the chemistry but also the rock texture has a strong influence. So the 4 lithological units should show different behaviour. Why do you derive one relation for all 4 lithological units?

I think that it would better to derive for relations of each lithological unit separately instead of single relations for all lithological units.

c.) Relations of rock properties (eq. 15, 16, ...)
What is the physical/petrophysical ansatz for the chosen form of the relations?
For example equation (16) is a sum of two exponential functions. Is this an arbitrary choice?

At equation (23) and (24) a Taylor series expansion is used. Taylor series expansion are only valid for small range around the anker point. Also the higher order terms (order >= 3) are neglected. Is this approximation valid for the P'-range of 0 - 150 MPa? How good is the fit?

d.) Figures of rock properties (Figure 4, 6, 7)
It is hard to assess the fit of the relations to the data in most of the figure.
All data are displayed leading to overcrowded plots as figure 6a and 6b. It is impossible to say whether the 4 colored lines fit the relevant data or not.
I suggest to create separate graphs for each subset/lithological unit.

In figure 4b I would plot data of dry and saturated rocks in different (sub)plots.

e.) Units of relation coefficients p1 - p17.
All coefficients are lacking of their unit.

For example p4 and p6 should have the unit of $Pa$ and p5 and p7 of $1/Pa$.

f.) Ratio between Kd and Gd
Page 11, line 1ff: References are presented suggesting a Kd/Gd-ratio of 1.7 or higher.
The authors decided to use a Kd/Gd-ratio of 1.5. Why is this value used?
Please explain in more detail or present other references to support a ratio of 1.5.

g.) Length of fracture segments
Would it be possible to include a histogram of the length of the fracture segments?
Maybe in figure 8?
Otherwise the claim that $l_d$ is similar to the half-length of most of the segments is not
supported.

h.)Amplitude and value of characteristic frequency
$A_C$ and $A_D$ (Page 18, line 34), $f_{c,C}$ and $F_{c,D}$ ((Page 19, line 1) are not visible in Figure
12.
How do you know that the inequalities are valid?
Those frequencies are outside to plotting range.

Minor comments:
- Page 2, line 7: replace wavespeed by velocity.
- Page 2, line 15 : replace wavespeed by velocity.
- page 5, line 8: replace incompressibility by bulk modulus.
- page 5, line 13f: "rock domain" and "fracture domain" are not used in Figure 1b.
Relabel the figure 1b.
- Page 10, line 1 : replace wavespeed by velocity.
- Page 10, line 26 : replace wavespeed by velocity.

[Figure]

- Page 11, line 1 : replace wavespeed by velocity.
- Page 11, line 15: Do you refer to figure 1b or 1c?
- Page 11, line 19: Is $rho_b$ part of the numerator or of the denominator? confusing. Better: $\rho_s = \rho_b/(1 - \Phi)$.
- Page 12, line 16: Is this formula correct? "eh" is occurring on both sides of the formula.
- Page 14, line 22: missing bracket ")".
- page 17, line 5: replace incompressibility by bulk modulus.
- page 20, line 32: replace incompressibility by bulk modulus.
- page 31, figure 1c: There is no reference to figure 1c in the text.

---

## Referee Comment (RC2) · Anonymous Referee #2 · 29 Dec 2016

Review of Grab et al: "Numerical modeling of fluid effects on seismic properties of fractured magmatic geothermal reservoirs " Submitted to Solid Earth Discussions

This paper concerns the influence of fluid-bearing fractures in magmatic and volcanic rocks, for the purpose of using seismic properties to characterize geothermal reservoirs. In particular it deals with the dispersion of seismic velocities (expressed through the P wave modulus and shear modulus) and the attenuation of the P and S waves ($1/Q_p$ and $1/Q_s$), using poroelastic numerical modelling based on Biot's theory. This approach is still fairly uncommon in standard rock physics applications, but has certainly become more common in the past 5 to 10 years, and is very powerful. In a fractured reservoir the seismic response is severely affected by the fluid flow, which

impacts the velocities and attenuation as a function of frequency in particular. Four different igneous rock types are tested, with a fairly large range in values of isotropic elastic constants, density, and porosity (and permeability). Fractures are incorporated into the 2D model, based on a geological study by Gudmundsson et al. (2002).

The paper is valuable and interesting; it is very well written, and easy to follow. I therefore recommend its publication with minor revisions. The abstract is clear and descriptive of the paper. The figures are generally well prepared and easy to follow (legend in Fig. 7c, d could be improved through). The language can be improved in some places (for example when using words like "what" and "why" after commas in sentences, where the word "which" should be used).

I have three general, but fairly minor, comments that the authors may want to address in the discussion of the paper: 1. The authors have a collected a large set of laboratory data, which is very good. However, it is not so clear to which confining pressures the laboratory experiments were made, and how the elastic moduli at zero confining pressure as well as the first and second velocity pressure derivatives were obtained. Most lab experiments have probably been carried out to confining pressures where the micro-cracks are not completely closed during pressurization (many 100's of MPa or even >1 GPa), which would make it difficult to obtain completely crack-free zero pressure elastic moduli. However, this is a secondary point when considering the effect of fluids in fractures on the seismic properties.

2. I think it is somewhat difficult to say how applicable these results are to an actual seismic data set and specific geological case. Since the igneous rocks can be rather heterogeneous. Therefore I view the contribution of Grab et al. as an exploratory study that demonstrates the importance of fluid effects on the seismic properties in a fractured rock, in a general sense. It would be nice for the future if the technique can be tested in situ, maybe in a borehole study

3. Although the results in the study builds in part on laboratory data, there are a lot of

numerical relationships and assumptions established and used in the paper, from the effective medium theory and the poroelastic theory. It is therefore difficult to appreciate how good the final model is in terms of testing it against real experiments or a laboratory case.

---

## Author Comment (AC1) · 26 Jan 2017

Dear Reviewers, We thank for the time and interest you invested for evaluating our manuscript. In the attached document, we respond to your questions and comments, and document the changes we made in order to improve the manuscript thanks to your valued inputs.

Best Regards,

Melchior Grab and Co-Authors

Please also note the supplement to this comment:

[Figure]

http://www.solid-earth-discuss.net/se-2016-144/se-2016-144-AC1-supplement.zip